# Synergistic effect of tumor chemo-immunotherapy induced by leukocyte-hitchhiking thermal-sensitive micelles

Jing Qi[1,4], Feiyang Jin[1,4], Yuchan You[1], Yan Du[1], Di Liu[1], Xiaoling Xu[1✉], Jun Wang[1], Luwen Zhu[1], Minjiang Chen[1], Gaofeng Shu[2], Liming Wu[3✉], Jiansong Ji[2✉] & Yongzhong Du[1✉]

Some specific chemotherapeutic drugs are able to enhance tumor immunogenicity and facilitate antitumor immunity by inducing immunogenic cell death (ICD). However, tumor immunosuppression induced by the adenosine pathway hampers this effect. In this study, E-selectin-modified thermal-sensitive micelles are designed to co-deliver a chemotherapeutic drug (doxorubicin, DOX) and an A2A adenosine receptor antagonist (SCH 58261), which simultaneously exhibit chemo-immunotherapeutic effects when applied with microwave irradiation. After intravenous injection, the fabricated micelles effectively adhere to the surface of leukocytes in peripheral blood mediated by E-selectin, and thereby hitchhiking with leukocytes to achieve a higher accumulation at the tumor site. Further, local microwave irradiation is applied to induce hyperthermia and accelerates the release rate of drugs from micelles. Rapidly released DOX induces tumor ICD and elicits tumor-specific immunity, while SCH 58261 alleviates immunosuppression caused by the adenosine pathway, further enhancing DOX-induced antitumor immunity. In conclusion, this study presents a strategy to increase the tumor accumulation of drugs by hitchhiking with leukocytes, and the synergistic strategy of chemo-immunotherapy not only effectively arrested primary tumor growth, but also exhibited superior effects in terms of antimetastasis, antirecurrence and antirechallenge.

[1] Institute of Pharmaceutics, College of Pharmaceutical Sciences, Zhejiang University, Hangzhou, China. [2] Key Laboratory of Imaging Diagnosis and Minimally Invasive Intervention Research, Lishui Hospital of Zhejiang University, Lishui, China. [3] Department of Hepatobiliary and Pancreatic Surgery, The First Affiliated Hospital, Zhejiang University School of Medicine, Hangzhou, China. [4] These authors contributed equally: Jing Qi, Feiyang Jin. ✉email: ziyao1988@zju.edu.cn; wlm@zju.edu.cn; lschrjjs@163.com; duyongzhong@zju.edu.cn

everal chemotherapeutic drugs, especially anthracyclines, have been repurposed to provoke antitumor immune responses by inducing immunogenic cell death (ICD) in addition to direct tumor-killing effects[1]. Tumor ICD is accompanied by the release of damage-associated molecular patterns (DAMPs), including the exposure of calreticulin (CRT), secretion of adenosine triphosphate (ATP), and release of high mobility group protein B1 (HMGB1)[2–6]. These DAMPs have been identified to facilitate dendritic cell (DC) maturation and antigen presentation to naive T cells[4,7]. Subsequently, the activation of T cells leads to the recruitment of cytotoxic T cells (CTLs) to the tumor site, thereby promoting tumor-specific cellular immunity, which can further enhance antitumor effects of chemotherapeutic agents[8,9].

Despite the ICD induction and immune response initiation of these select chemotherapeutic drugs, there remain challenges. Tumor cells can release large amounts of ATP during ICD induced by chemotherapeutic drugs, subsequently metabolized to adenosine (ADO, a potent immunosuppressor) by ectonucleotidases, such as CD39 and CD73[10]. The engagement of ADO and the ADO 2A receptors (A2AR, an immune checkpoint) on various immune cell surfaces hampers the immune reaction toward tumor cells, further exacerbating tumor immunosuppression[11–13]. Therefore, the paradoxes between ICD-induced antitumor immunity and ADO-mediated immunosuppression remain a formidable challenge. Fortunately, preclinical studies targeting the adenosinergic pathway have gained much attention for their clinical potential in overcoming tumor-induced immunosuppression. Blockade of the ectonucleotidases that generate ADO, or the A2AR that mediates adenosinergic signals in immune cells, will greatly contribute to restraining tumor growth and metastasis[14–18]. This suggests the possible benefits of utilizing ADO-related therapeutic approaches in combination with chemotherapeutic drugs with ICD induction ability. In particular, antagonists of A2AR are just occurring to be deployed into oncology, which can block the interaction between ADO and A2AR, thereby alleviating tumor immunosuppression and facilitating the antitumor immune response[19,20]. It is worth noting that A2AR is widely distributed on a variety of immune cells and is a ubiquitous immune checkpoint, which holds promise for addressing the low response rate of PD-1/PD-L1 blockade therapies[18]. Therefore, the combined application of chemotherapeutic drugs and A2AR antagonists may amplify antitumor efficacy.

However, both chemotherapeutic drugs and A2AR antagonists have limited tumor-targeting ability after intravenous administration, which often induces undesirable adverse effects and unsatisfactory efficacy. Smart nanoparticle drug delivery system is an effective way to alter biodistribution of drugs and achieve spatiotemporally controlled drug release, which is beneficial for improving treatment safety and efficacy[8,21–23]. Significantly, thermal-sensitive drug delivery system has attracted much attention; hyperthermia stimuli at the tumor site can accelerate the drug release from nanoparticles to achieve precise therapy, and on the other hand, hyperthermia itself can also suppress tumor growth[24,25]. Despite these advantages, delivering nanoparticle platforms in patients with advanced forms of cancer remains a challenge. Only a fraction of all drug-loaded nanoparticles can reach the tumor site, while the vast majority of nanoparticles are cleared by the reticuloendothelial system (RES), and the clinical translation of the EPR effect from animal models to humans has been proven to be challenging[26]. In addition, elevated fluid pressures and the lack of well-defined vasculature also hinder the application of nanoparticles in tumor therapy[27–29].

A strategy that potentially addresses the challenges listed above and optimizes biodistribution in a highly specific manner involves the use of circulating cells to mediate the transport of drug-loaded nanoparticles[30–32]. Several kinds of circulating cells, including neutrophils[33], T cells[34–36], NK cells[37], and erythrocytes[38], have been successfully applied in this kind of hitchhiking strategy. Specifically, leukocytes, which share similar migration patterns to tumor cells in blood and tissues[39], can also be utilized to carry drug-loaded nanoparticles and pass challenging biological barriers to accumulate in tumor sites[40,41].

In this work, E-selectin-modified thermal-sensitive micelles (ES-DSM) are fabricated, which co-load with the chemotherapeutic drug doxorubicin (DOX) and the A2AR antagonist SCH 58261 (hereafter referred to as SCH). After intravenous administration, the ES-DSM can hitch a ride on leukocytes mediated by E-selectin to across biological barriers and achieve increased tumor accumulation. Subsequently, local microwave stimulation is applied to induce hyperthermia and accelerates the release rate of drugs from nanoparticles. Rapidly released DOX not only directly kills tumor cells but also improves tumor immunogenicity by inducing ICD. The maturation and antigen presentation of DCs are facilitated, and the further tumor-specific T-cell immunity is elicited. On the other hand, released SCH prevents the engagement of ADO with A2AR on the surface of various immune cells, which relieves the immunosuppression phenomenon and further enhances DOX-induced tumor-specific cellular immunity (Fig. 1). Consequently, considerably enhanced antitumor efficacy can be achieved via the synergistic effect of chemo-immunotherapy.

## Results
**Characterization of NTA-PEG-p-(AAm-co-AN).** First, the amphiphilic polymer, nitrilotriacetic acid-PEG-poly-(acrylamide-co-acrylonitrile) (NTA-PEG-p-(AAm-co-AN)) (Fig. 2a), was synthesized according to Supplementary Fig. 1. The chemical structure of the polymers was confirmed by $^1$H-NMR spectra as shown in Fig. 2b and Supplementary Fig. 2. The molecular weights of p-(AAm-co-AN) and PEG-p-(AAm-co-AN) were measured as 10.9 and 14.3 kDa, respectively. To evaluate the thermal sensitivity of the polymer, turbidity measurements were performed to determine the upper critical solution temperature (UCST) of p-(AAm-co-AN). As shown in Fig. 2c, the transmittance of the polymer solution increased from 4 to 43 °C and became constant above 43 °C, which confirmed that the UCST value of the polymer was 43 °C. Further, synthesized NTA-PEG-p-(AAm-co-AN) was found to self-assemble into micelles in aqueous solution at ambient temperature, and the critical micelle concentration (CMC) was determined to be 33.2 µg/mL (Fig. 2d). Importantly, blank micelles that self-assembled from NTA-PEG-p-(AAm-co-AN) were proven to be thermal-sensitive. As exhibited in Fig. 2e, blank micelles presented regular and uniform spherical morphologies at both 25 and 37 °C, but irregular shapes at 43 and 50 °C, supporting the stability of blank micelles at physiological temperature (37 °C) as well as their destruction under hyperthermic condition (43 °C). NTA in the polymer was used to chelate $Ni^{2+}$ to afford Ni-NTA, which could further efficiently bind to the His-tag of recombinant E-selectin, thereby introducing E-selectin onto the surface of micelles. The chelating ability of NTA-PEG-p-(AAm-co-AN) to $Ni^{2+}$ was demonstrated by ICP-MS, and the result showed that 0.96 mol of $Ni^{2+}$ could be chelated per mole of the polymer.

**Characterization of E-selectin-modified DOX and SCH co-loaded micelles (ES-DSM).** Subsequently, DOX and SCH co-loaded micelles (DSM) were prepared with feed ratios of DOX and SCH of 4% and 1%, respectively. The encapsulation efficiency and drug loading of DOX were $92.9 \pm 0.61\%$ and $2.7 \pm 0.01\%$, respectively, while those of SCH were $41.8 \pm 0.97\%$ and $0.41 \pm 0.005\%$,

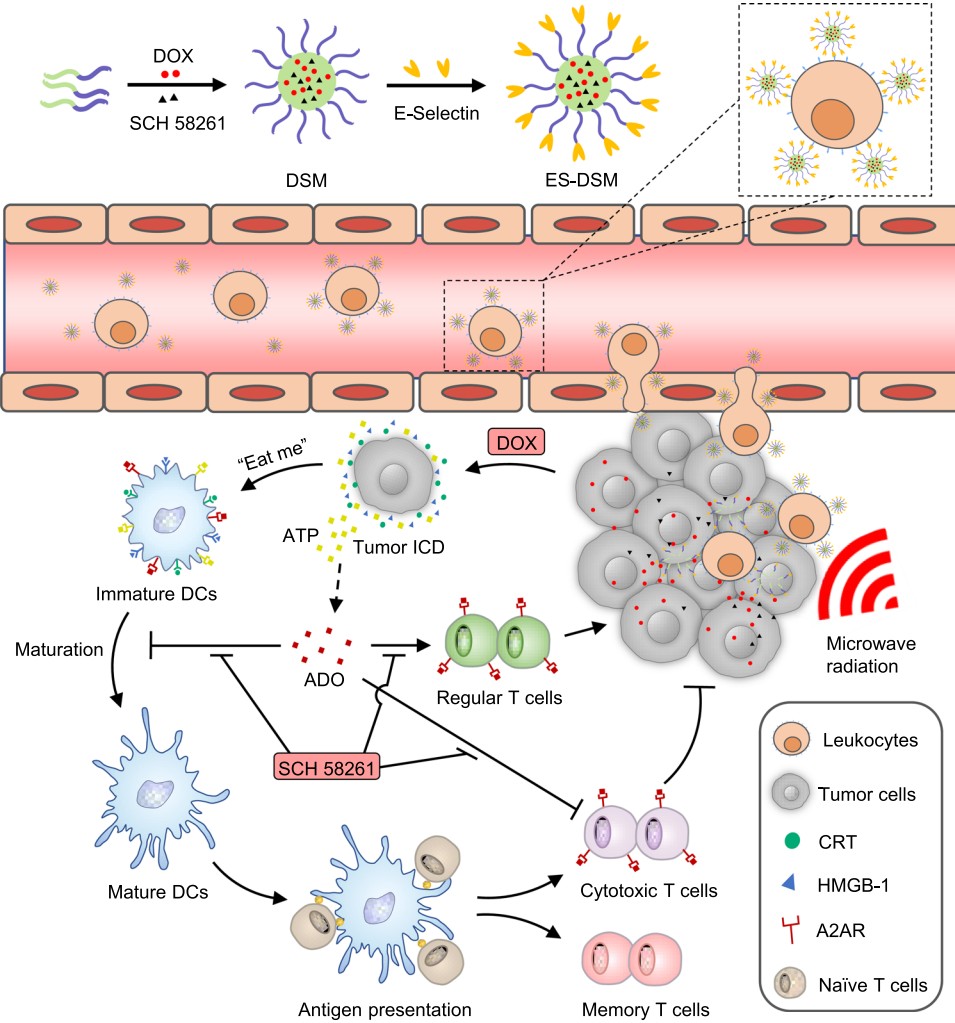

**Fig. 1 Schematic depiction of the fabrication of ES-DSM and the synergistic effect of chemo-immuno-microwave hyperthermia therapy of ES-DSM delivered by leukocytes.** ES-DSM: E-selectin-modified drug-loaded micelles.

respectively. Further, E-selectin was introduced onto the micelle surface to obtain ES-DSM. As shown in Supplementary Fig. 3, as E-selectin modifications increased, the particle size of ES-DSM increased, while the potential decreased. ES-DSM applied in this study was prepared by adding 2 μg/mL E-selectin into a solution of 1 mg/mL polymer. Figure 2f showed that the particle size and potential of DSM were $164.0 \pm 7.0$ nm and $3.93 \pm 0.05$ mV, respectively. However, when E-selectin was introduced onto micelles to form ES-DSM, the particle size increased to $247.7 \pm 15.6$ nm while the potential decreased to $-1.2 \pm 0.09$ mV, which further proved the successful preparation of ES-DSM. The spherical morphology of ES-DSM was also observed by TEM (Fig. 2g).

Further, the thermal sensitivity of micelles was investigated by determining particle sizes at different temperatures. As presented in Fig. 2h, the size of blank micelles remained below 100 nm at 5–37 °C, while it was almost undetectable at 43 °C and above, which was consistent with the TEM results in Fig. 2e. Importantly, the sizes of DSM and ES-DSM increased to more than 1000 nm when detected at 43 °C and above, which was due to the dissolution of the micelles under thermal conditions, and the insoluble drugs DOX and SCH were released immediately to form precipitates. Afterward, the thermal-sensitive in vitro drug release behavior of ES-DSM was evaluated by the dialysis method at 37 and 43 °C. As shown in Fig. 2i and j, under physiological condition (37 °C), the drug release rates were relatively slow, and ~40% and 50% of SCH and DOX were

released, respectively, within 48 h. However, under thermal condition (43 °C), the release rates of SCH and DOX were considerably accelerated and were similar to the profile of free drugs. The rapid drug release behavior of ES-DSM at 43 °C was the result of micelle disintegration.

Subsequently, the specific recognition ability of ES-DSM to leukocytes was evaluated. Both DSM and ES-DSM were demonstrated to be biocompatible with leukocytes and had no significant impact on cell viability, chemotaxis, and penetration ability (Supplementary Fig. 4). At different times after the intravenous injection of DSM or ES-DSM, leukocytes were isolated by the mouse peripheral blood leukocyte separation kit according to manufacturer's instructions, and the fluorescence intensity of DOX was detected by flow cytometry. Figure 2k and l showed that the fluorescence intensity of leukocytes exhibited a negligible change within 24 h after DSM injection but was significantly enhanced after ES-DSM injection, and ~30% of leukocytes were DOX positive at 24 h post-injection. In addition, leukocytes were isolated 24 h after injection and observed by confocal microscopy, which demonstrated that ES-DSM adhered to the surface of leukocytes (Fig. 2m and Supplementary Fig. 5). Taken together, in contrast to DSM, ES-DSM presented an efficient leukocyte targeting ability and adhered to the surface of leukocytes, further emphasizing the important role of E-selectin in the hitchhiking of micelles to leukocytes.

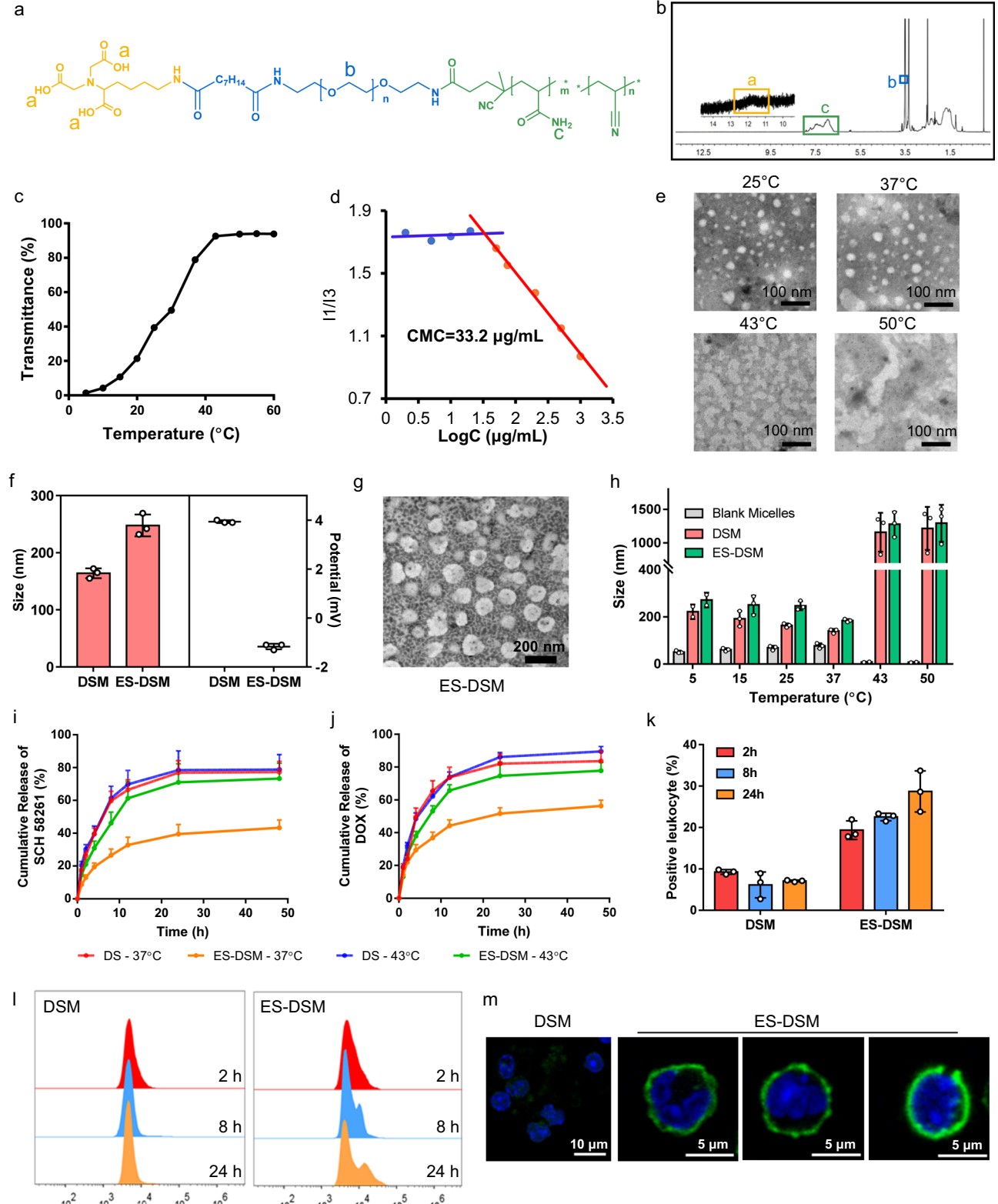

**Cellular drug release, cytotoxicity, and ICD induction ability of ES-DSM supplemented with hyperthermia.** Next, the thermal-sensitive drug release behavior at the cellular level was investigated by confocal microscopy. First, Nile red was used as the model drug to prepare Nile red-loaded micelles. When 4T1 cells were exposed to Nile red-loaded micelles and treated with

hyperthermia (+), Nile red was released rapidly and bound with the intracellular lipid membrane, and fluorescence was observed, which was similar to free Nile red. However, cells without hyperthermia (−) exhibited weaker fluorescence intensity because the drug was not released (Fig. 3a and Supplementary Fig. 6). In addition, when 4T1 cells were exposed to DOX-loaded micelles,

**Fig. 2 Characterization, thermal sensitivity, and leukocyte targeting ability of ES-DSM. a** Chemical structure of NTA-PEG-p-(AAm-co-AN). **b** $^1$H-NMR spectra of NTA-PEG-p-(AAm-co-AN) and the characteristic peaks were marked by rectangles. -O-CH$_2$-CH$_2$-: 3.6 ppm, -CONH$_2$: 6.7–7.9 ppm, -COOH: 11.5–12.5 ppm. **c** The transmittance of p-(AAm-co-AN) aqueous solution (2 mg/mL) at different temperatures. **d** Critical micelle concentration (CMC) of NTA-PEG-p-(AAm-co-AN). **e** TEM images of blank micelles at different temperatures. **f** Hydrodynamic size and zeta potential of DSM and ES-DSM ($n = 3$ independent experiments). **g** TEM image of ES-DSM at 25 °C. **h** Hydrodynamic sizes of blank micelles, DSM and ES-DSM after incubation at different temperatures for 10 min ($n = 3$ independent experiments). The thermal-sensitive in vitro release behavior of **i** SCH and **j** DOX from ES-DSM at 37 or 43 °C ($n = 3$ independent experiments). **k–l** Flow cytometry analysis of leukocytes fluorescence in blood after the intravenous injection of DSM or ES-DSM for different times ($n = 3$ mice). Positive percentage of leukocytes in (**k**) was calculated based on (**l**). **m** Confocal microscopy images of leukocytes 24 h after the intravenous injection of DSM or ES-DSM. Leukocytes have nuclear morphology characteristic of neutrophils (right), monocytes (center), and lymphocytes (right). DSM: DOX and SCH co-loaded micelles; ES-DSM: E-selectin-modified co-loaded micelles. Data are presented as mean values ± SEM, and the mean value is the average of three independent experiments. Unpaired two-tailed $T$ test was performed in (**f**), (**h**), (**i**), (**j**), and (**k**). The experiments in (**e**), (**g**), and (**m**) were repeated independently for three times with similar results. Source data are provided as a Source data file.

after being treated with hyperthermia (+), DOX was liberated and obviously entered the nucleus, which was similar to free DOX. When treated without hyperthermia (−), DOX resided in micelles and was therefore mainly distributed in the cytoplasm (Fig. 3b). These results indicated the thermal-sensitive nature of drug-loaded micelles at the cellular level.

Then, the cytotoxicity of free DOX and SCH (DS), DSM, and ES-DSM was assessed. Initially, the biocompatibility of blank micelles was confirmed, and hyperthermia treatment did not affect 4T1 cell viability (Supplementary Fig. 7). After exposure to DS, DSM, or ES-DSM with or without hyperthermia, 4T1 cell viability was measured by MTT assay. In Fig. 3c and d, there was no significant difference in cytotoxicity between the groups of DS supplemented with or without hyperthermia (IC$_{50}$ values were 8.50 and 8.45 μM, respectively). However, compared to the DSM and ES-DSM treated groups (IC$_{50}$ values were 30.70 and 29.35 μM, respectively), the hyperthermia-treated groups exhibited higher cytotoxicity (IC$_{50}$ values were 11.25 and 10.50 μM, respectively), which was similar to the toxicity of free drugs (DS). The reason for this difference was that the drugs could be released immediately from micelles under the thermal condition to execute their tumor cell killing function. Importantly, the modification of E-selectin exhibited negligible interference on cytotoxicity of drug-loaded micelles. In addition, 4T1 cell apoptosis induced by different treatments was detected by flow cytometry. As displayed in Fig. 3e and f, the DSM and ES-DSM treated groups supplemented with hyperthermia presented more severe early and late apoptosis than the unheated groups. All of these results indicated that the drug-loaded micelles applied with hyperthermia exhibited more effective antitumor effect than the unheated groups, which was attributed to the thermal-sensitive release behavior of drugs from micelles.

In addition, the ICD induction ability of drug-loaded micelles was analyzed. DOX can efficiently induce ICD in tumors, which is accompanied by the exposure of CRT, secretion of ATP, and release of HMGB1 (Fig. 3g). Therefore, we tested whether enhanced exposure of CRT, ATP, and HMGB1 was observed when 4T1 cells were incubated with different agents with or without hyperthermia. As displayed in Fig. 3h and i, free DOX could significantly increase the CRT exposure on tumor cells and hyperthermia had no significant effect on this efficacy. However, DSM or ES-DSM applied without hyperthermia induced less CRT exposure, which was due to the slow release of DOX from micelles under physiological conditions. Importantly, the exposure level of CRT increased when DSM or ES-DSM was combined with hyperthermia, which was similar to the free drugs. The exposure levels of ATP and HMGB1 also exhibited similar results with CRT (Fig. 3j and k).

**Maturation of DCs in the binary co-incubation system.** During the ICD process of tumor cells, CRT is overexpressed and provides an "eat-me" signal for dendritic cell uptake[4,5], while released

HMGB1 and ATP serve as adjuvant stimuli for dendritic cell maturation (Fig. 4a)[6]. Therefore, after 4T1 cells were exposed to different agents with or without hyperthermia and incubated for 24 h, immature DCs were added to co-incubate for another 48 h, and biomarkers of mature DCs (CD80, CD86, and MHC II) were analyzed by flow cytometry. As shown in Fig. 4b, c, f, g and Supplementary Fig. 8, when 4T1 cells were pretreated with DSM or ES-DSM and hyperthermia, they promoted the maturation of DCs. The expression of CD80, CD86, and MHC II was similar to that in the free drug (DS) treated groups but significantly higher than that in the unheated DSM or ES-DSM treated groups. Moreover, immunologic factors secreted by DCs were monitored by ELISA kits. Figure 4h–j demonstrated that levels of IL-12p70 (a DC-secreted immune-related cytokine) and IL-6 in the suspension of the co-incubation system increased while IL-10 decreased when DSM or ES-DSM were applied in combination with hyperthermia, which was consistent with the DS treated groups. These results further supported the thermal-sensitive property of the drug-loaded micelles and that the ICD of tumor cells facilitated DC maturation.

It is worth noting that ADO in the tumor environment can bind to A2AR on the DC surface, thereby inhibiting DC maturation and antigen presentation. SCH serves as an antagonist to block the interaction between ADO and A2AR at the DC surface, further relieving the immunosuppression of DCs (Fig. 4a). To verify the effect of SCH on the immune response, 1 μM of NECA (an analog of ADO) was added to the co-incubation system to simulate the tumor microenvironment[42], and then DC maturation was evaluated. As displayed in Fig. 4d, e, k, l and Supplementary Fig. 9, when only DOX (groups of D, DM, and ES-DM with hyperthermia) was in the co-incubation system, the expression of CD80, CD86, and MHC II was lower than that of the groups containing both DOX and SCH (groups of DS, DSM, and ES-DSM with hyperthermia), which also exhibited more secretion of IL-12p70 and IL-6 but less IL-10 (Fig. 4m–o). These results showed that the presence of NECA arrested the maturation of DCs, but SCH relieved this phenomenon by blocking the interaction between NECA and A2AR.

**Activation of T cells in the ternary co-incubation system.** Mature DCs facilitated by tumor ICD can present antigens to naive T cells, further promote their differentiation into cytotoxic T cells (CTLs) or regulatory T cells (Tregs), and finally elicit T-cell immune responses (Fig. 5a). Therefore, a ternary co-incubation system of 4T1 cells (which had been pretreated with different agents with or without hyperthermia), immature DCs, and splenic lymphocytes was constructed and cultured for 48 h. Subsequently, the proliferation of CD3$^+$CD4$^+$ and CD3$^+$CD8$^+$ T cells was analyzed. As exhibited in Fig. 5b, d, e, when 4T1 cells were pretreated with DSM or ES-DSM in combination with

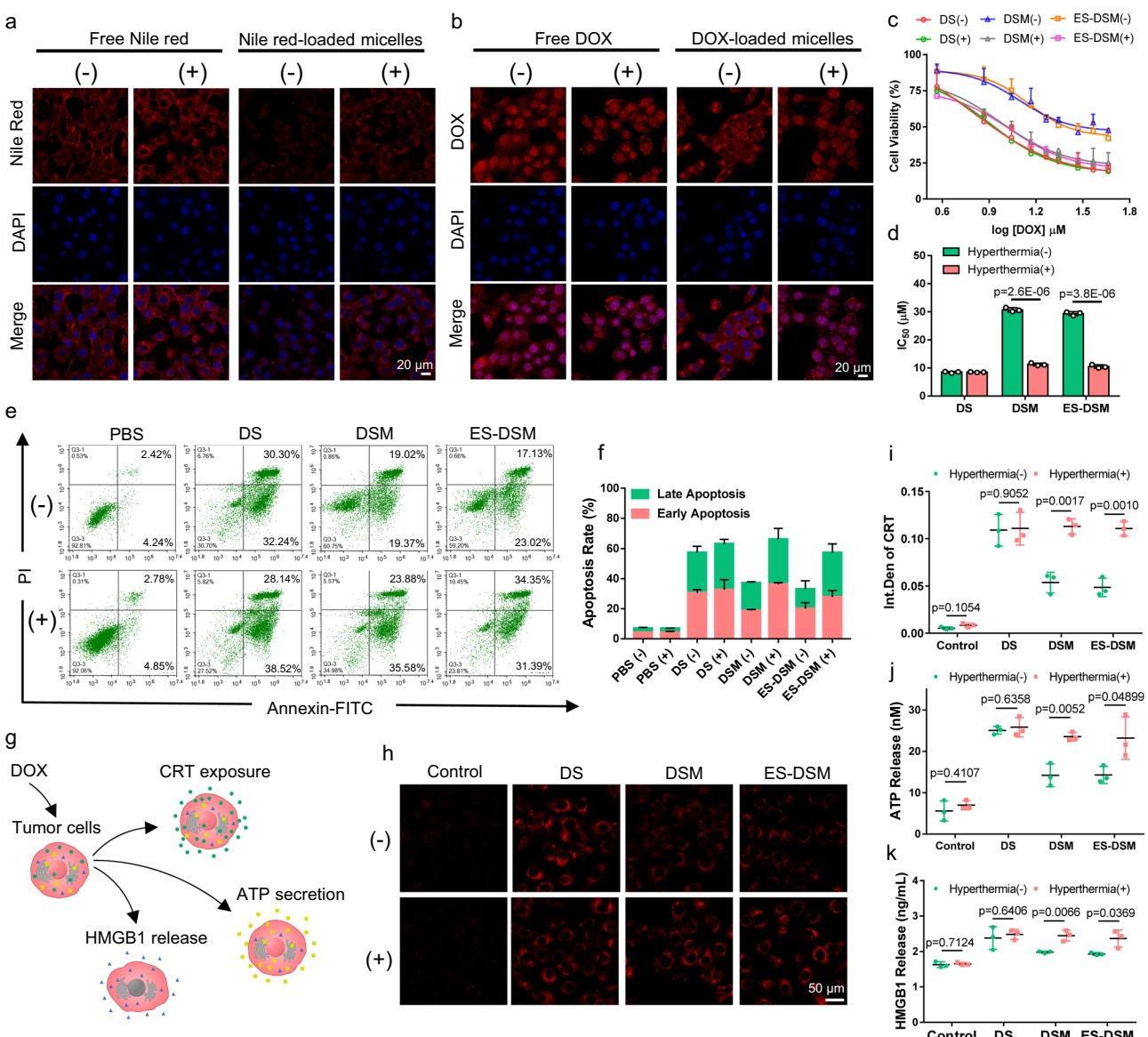

**Fig. 3 Thermal-sensitive drug release at the cellular level and cytotoxicity of DS, DSM, and ES-DSM.** Confocal microscopy images of 4T1 cells exposed to **a** free Nile red or Nile red-loaded micelles, and **b** free DOX or DOX-loaded micelles and treated with (+) or without (−) hyperthermia. **c** Variations in 4T1 cell viability after exposure to DS, DSM, or ES-DSM for 48 h as a function of the concentration of DOX with (+) or without (−) hyperthermia ($n = 3$ independent experiments). **d** IC$_{50}$ values of different treatments were calculated based on (**c**). **e** The apoptosis results of 4T1 cells after different treatments for 24 h with (+) or without (−) hyperthermia detected by flow cytometry ($n = 3$ independent experiments). **f** The apoptosis rate of 4T1 cells was calculated based on (**e**). **g** Schematic showing that DOX-induced ICD in 4T1 cells accompanied by CRT exposure, ATP secretion, and HMGB1 release. **h** CRT exposure of 4T1 cells after different treatments was observed by confocal microscopy. **i** Semi-quantitative analysis of **h** using Image J ($n = 3$ independent experiments). **j** ATP secretion was detected by luciferase conversion assay ($n = 3$ independent experiments). **k** HMGB1 release was measured by ELISA kits ($n = 3$ independent experiments). DS: free DOX and SCH. DSM: DOX and SCH co-loaded micelles; ES-DSM: E-selectin-modified co-loaded micelles. Data are presented as mean values ± SEM, and the mean value is the average of three independent experiments. Unpaired two-tailed T test was performed in (**d**), (**i**), (**j**), and (**k**). The experiments in (**a**), (**b**), and (**h**) were repeated independently for three times with similar results. Source data are provided as a Source data file.

hyperthermia, both CD3$^+$CD4$^+$ and CD3$^+$CD8$^+$ T cells in the co-incubation system proliferated significantly and were more abundant than those in unheated groups. The negligible difference between the drug-loaded micelles with hyperthermia and free drugs treated groups suggested that the thermal-sensitive drug release behavior enabled micelles to execute the efficient antitumor effect. Further, CD4$^+$Foxp3$^+$ T cells, known as regulatory T cells (Tregs), which can hamper effective antitumor immunity, were obviously decreased when DSM and ES-DSM were applied with hyperthermia, suggesting that tumor ICD effectively stimulated T-cell immunity and weakened the

immunosuppressive effect of Tregs (Supplementary Fig. 10). Besides, the cytokines (TNF-α, IL-2, and IFN-γ) secreted by lymphocytes in the co-incubation system treated with drug-loaded micelles with hyperthermia exhibited a trend similar to that of the free drug groups (Fig. 5f–h). These results proved that the 4T1 cell ICD induced by thermal-sensitive drug-loaded micelles facilitated the antigen-presenting ability of DCs to naive T cells, further promoting their differentiation into CTLs rather than Tregs.

Importantly, ADO can interact with A2AR on the surface of T cells to inhibit the antitumor effect of CTLs and facilitate the

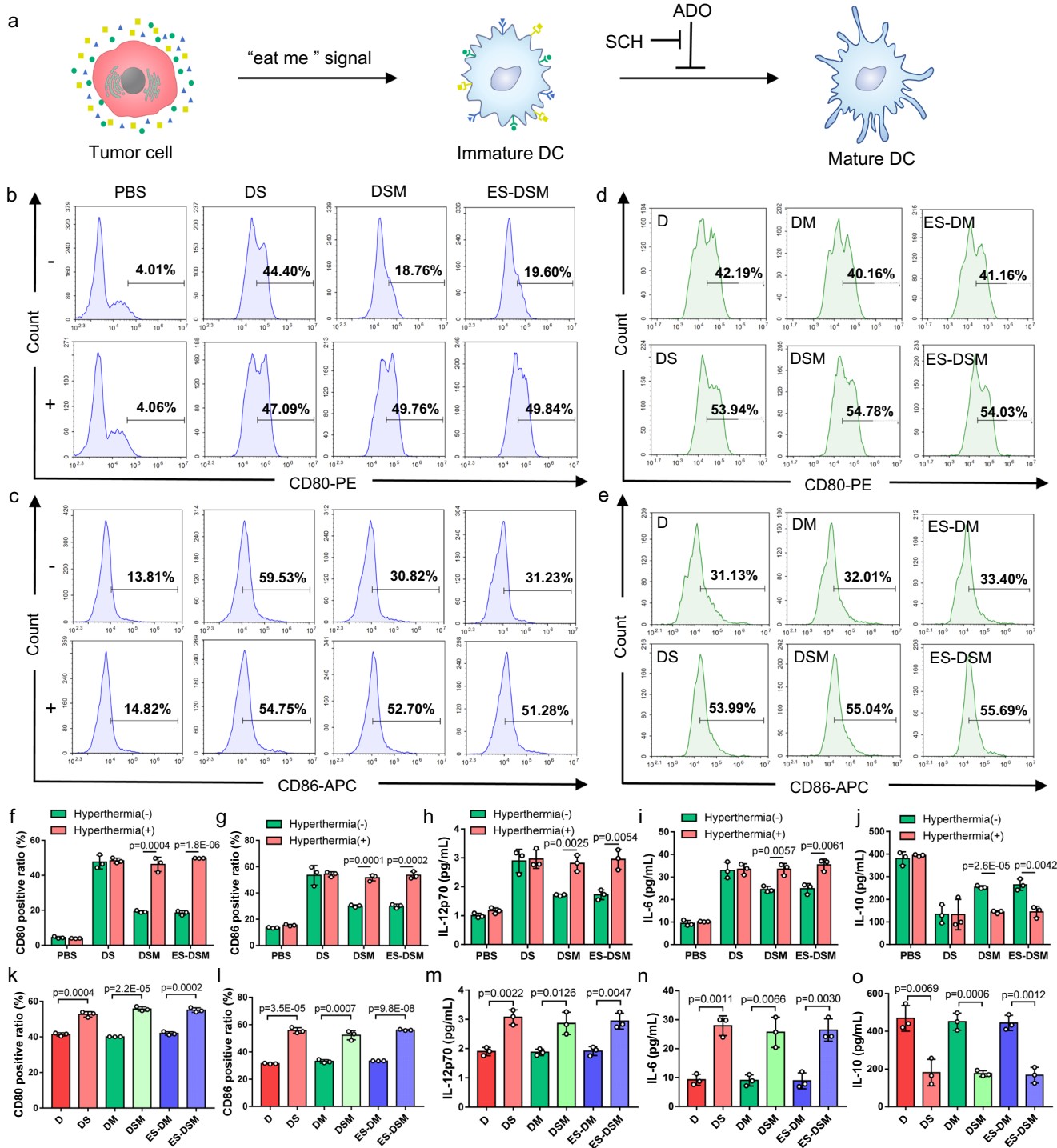

**Fig. 4 Analysis of DCs after co-incubating with pretreated tumor cells. a** Schematic of DC maturation facilitated by tumor ICD. ADO can inhibit this process by binding to A2AR on DCs, and SCH can block this interaction and relieve immunosuppression. Flow cytometry analysis of the expression of **b** CD80 and **c** CD86 on DCs after co-incubation with pretreated tumor cells, as well as **d** CD80 and **e** CD86 on DCs after co-incubation with pretreated tumor cells in the presence of NECA. Ratios of **f** CD80 and **g** CD86 positive DCs calculated based on (**b**) and (**c**), respectively ($n = 3$ independent experiments). **h** IL-12p70, **i** IL-6, and **j** IL-10 secreted by DCs in the co-incubation system after different treatments were detected by ELISA kits ($n = 3$ independent experiments). Ratios of **k** CD80 and **l** CD86 positive DCs calculated based on (**d**) and (**e**), respectively ($n = 3$ independent experiments). **m** IL-12p70, **n** IL-6, and **o** IL-10 secreted by DCs in the NECA-containing co-incubation system after different treatments were detected by ELISA kits ($n = 3$ independent experiments). ADO: adenosine; A2AR: A2A adenosine receptor; NECA: an analog of adenosine; DS: free DOX and SCH; DSM: DOX and SCH co-loaded micelles; ES-DSM: E-selectin-modified co-loaded micelles; D: free DOX; DM: DOX-loaded micelles; ES-DM: E-selectin-modified DOX-loaded micelles. Data are presented as mean values ± SEM, and the mean value is the average of three independent experiments. Unpaired two-tailed *T* test was performed in (**f**–**o**). Source data are provided as a Source data file.

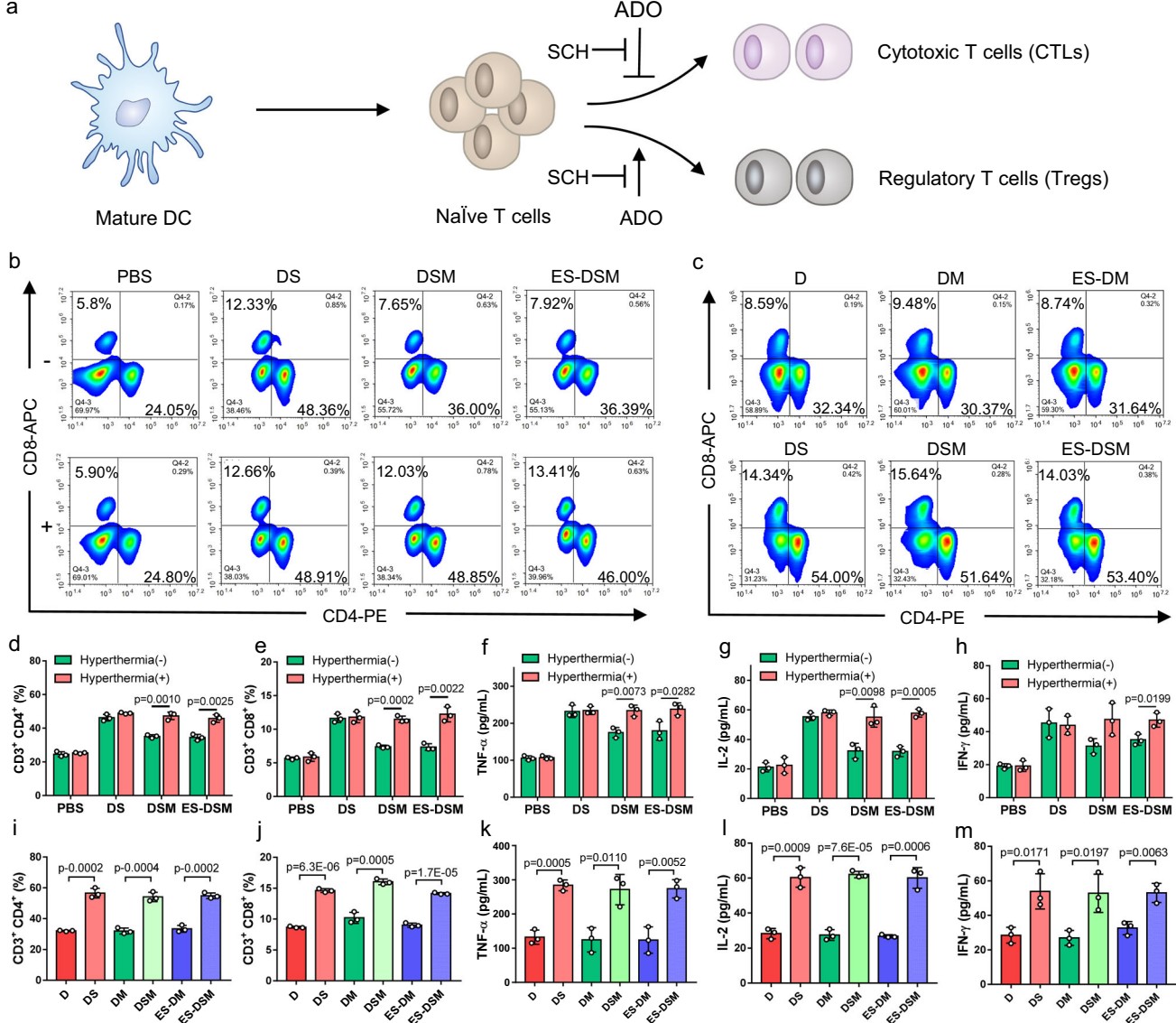

**Fig. 5 Analysis of T cells after co-incubating with pretreated tumor cells and DCs. a** Schematic of T-cell activation and differentiation facilitated by mature DCs. ADO can inhibit CTLs and promote Tregs by interacting with A2AR on the T-cell surface, and SCH can block this interaction and relieve immunosuppression. Flow cytometry analysis of percentages of $CD3^+CD4^+$ and $CD3^+CD8^+$ T cells **b** in the ternary co-incubation system and **c** in the ternary co-incubation system containing NECA. Ratios of **d** $CD3^+CD4^+$ and **e** $CD3^+CD8^+$ T cells calculated based on (**b**) ($n = 3$ independent experiments). **f** TNF-α, **g** IL-2, and **h** IFN-γ secreted by lymphocytes in the co-incubation system after different treatments were detected by ELISA kits ($n = 3$ independent experiments). Ratios of **i** $CD3^+CD4^+$ and **j** $CD3^+CD8^+$ T cells calculated based on **c** (n=3 independent experiments). **k** TNF-α, **l** IL-2, and **m** IFN-γ secreted by lymphocytes in the NECA-containing co-incubation system after different treatments were detected by ELISA kits ($n = 3$ independent experiments). ADO: adenosine; A2AR: A2A adenosine receptor; NECA: an analog of adenosine; DS: free DOX and SCH; DSM: DOX and SCH co-loaded micelles; ES-DSM: E-selectin-modified co-loaded micelles; D: free DOX; DM: DOX-loaded micelles; ES-DM: E-selectin-modified DOX-loaded micelles. Data are presented as mean values ± SEM, and the mean value is the average of three independent experiments. Unpaired two-tailed T test was performed in (**d-m**). Source data are provided as a Source data file.

immunosuppressive impact of Tregs. Fortunately, SCH can block the interaction between ADO and A2AR on the T-cell surface, thereby reversing the undesired immunosuppressive phenomenon (Fig. 5a). To verify this effect, 1 μM of NECA was added to the ternary co-incubation system, and the percentages of $CD3^+CD4^+$, $CD3^+CD8^+$, and $CD4^+$ $Foxp3^+$ T cells were detected. Figure 5c, i, j and Supplementary Fig. 11 showed that the application of SCH (groups of DS, DSM, and ES-DSM with hyperthermia) liberated T cells from the negative impact of NECA and promoted the proliferation of antitumor T cells. In addition, the levels of secreted cytokines (TNF-α, IL-2, and IFN-γ) also demonstrated the anti-immunosuppressive effect of SCH (Fig. 5k–m).

**In vivo antitumor efficacy of ES-DSM with microwave radiation (MW).** Next, the biodistribution of drug-loaded micelles was investigated in 4T1 tumor-bearing mice, and ICG was used as the model drug. ICG-loaded micelles with or without E-selectin modification were intravenously injected. As shown in Fig. 6a and Supplementary Fig. 12, ICG-loaded micelles with or without E-selectin modification could accumulate at the tumor site. However, E-selectin-modified micelles exhibited less liver accumulation and more tumor targeting at 24 h post-injection. Further, CD45 (a biomarker of leukocytes) in tumor sections was labeled and observed. As displayed in Fig. 6b, the fluorescence of ICG (red) and CD45 (green) overlapped obviously after the injection of E-selectin-modified ICG-loaded micelles, indicating that the

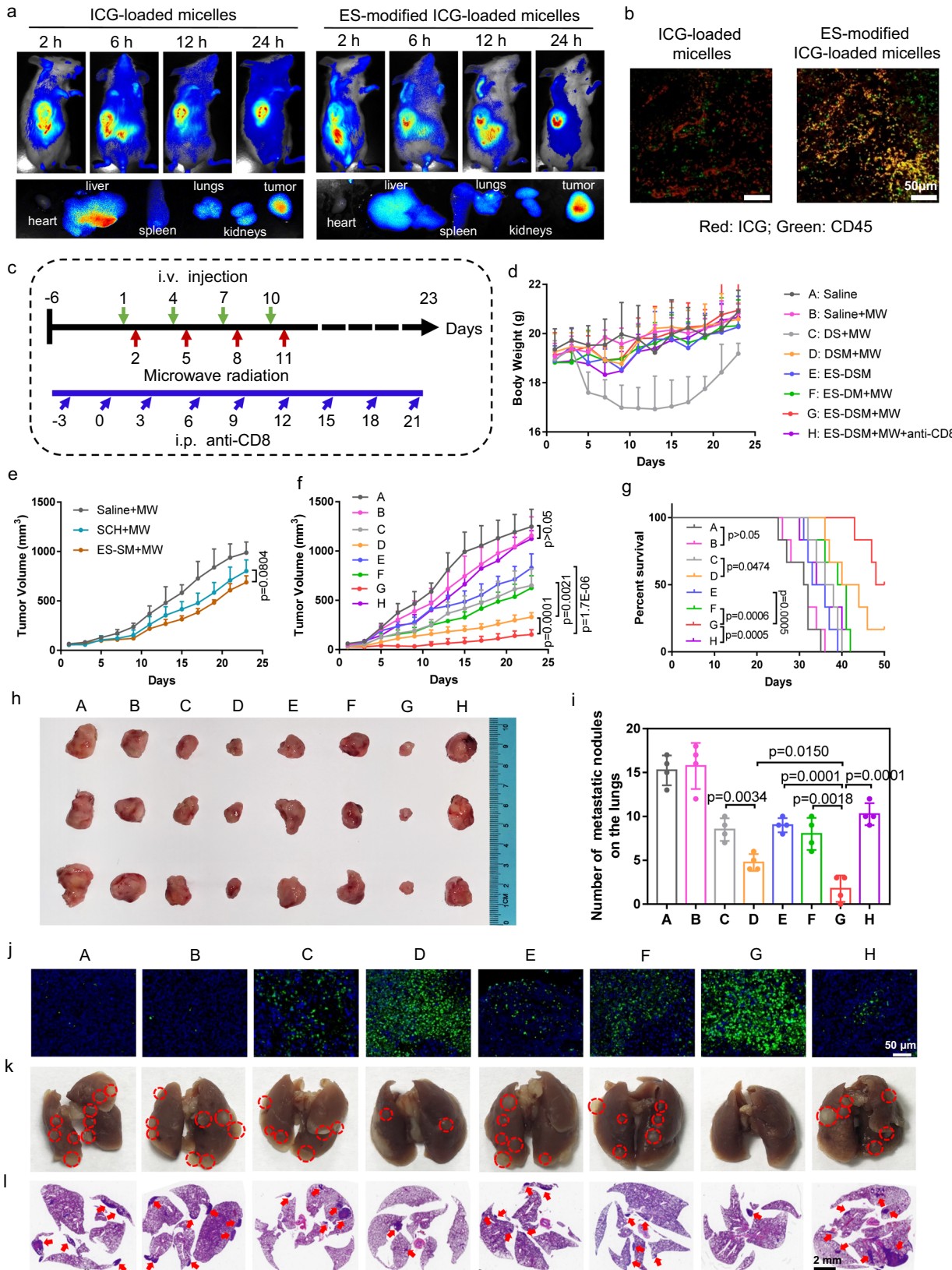

increase of micelles in tumors was benefited from hitching a ride on leukocytes.

Thereafter, the antitumor efficacy of DSM and ES-DSM was explored and the treatment regimen was displayed in Fig. 6c. Mice were intravenously (i.v.) injected with different agents every 3 days,

and in situ microwave thermotherapy was performed 24 h after i.v. injection, for 4 consecutive doses. In addition, to examine the effect of CD8$^+$ T cells on the antitumor immune response, an anti-CD8 antibody was intraperitoneally (i.p.) injected every 3 days to deplete CD8$^+$ T cells starting on day −3. The body weight of the free drug-

**Fig. 6 In vivo antitumor efficacy and the evaluation of pulmonary metastasis in 4T1 tumor-bearing models. a** Biodistribution of ICG-loaded micelles and E-selectin-modified ICG-loaded micelles in tumor-bearing mice within 24 h, and fluorescence images of tumors and major organs at 24 h after i.v. injection. ES refers to E-selectin. **b** Fluorescence images of ICG (red) and CD45 (green) in tumor tissues after the injection of ICG-loaded micelles or ES-modified ICG-loaded micelles. **c** Schematic of the treatment regimen. **d** Change curves of mice weights after various treatments ($n = 6$). **e, f** Curves showing tumor volumes of mice after various treatments ($n = 6$). **g** Survival curves of mice after various treatments ($n = 6$). **h** Representative photographs of harvested tumors after different treatments. **i** Number of metastatic tumor nodules on the lungs ($n = 4$). **j** Representative photographs of tumor tissues stained by TUNEL. **k** Representative photographs of lung tissues at the end of the observation period, and the metastatic tumor nodules were marked by red circles. **l** H&E staining of lung tissues, and the tumor areas were indicated by red arrows. DS: free DOX and SCH; DSM: DOX and SCH co-loaded micelles; ES-DSM: E-selectin-modified co-loaded micelles; ES-DM: E-selectin-modified DOX-loaded micelles; MW: microwave radiation (8 W, 30 min). Data are presented as mean values ± SEM. Unpaired two-tailed $T$ test was performed in (**e**), (**f**), (**i**), and Log-rank Mantel–Cox tests were performed in (**g**). The experiments in (**b**) and (**j**) showed similar results in three independent mice. Source data are provided as a Source data file.

treated group (DS + MW) decreased significantly compared to that of the other drug-loaded micelle groups, suggesting that the micelles reduced the side effects of free drugs (Fig. 6d). Changes in tumor volume were shown in Fig. 6e, f and Supplementary Fig. 13, and the photograph of tumor tissues at the end of the observation period was displayed in Fig. 6h. Compared with the group treated with saline (Saline), the application of microwave radiation (Saline + MW) exhibited negligible efficacy, and the tumor inhibition rate was ~7.2%. Free SCH plus microwave radiation (SCH + MW) exhibited poor efficacy, indicating the limited effect of SCH alone. Importantly, the E-selectin-modified SCH-loaded micelles and microwave hyperthermia (ES-SM + MW) showed no significant difference in comparison with the SCH + MW group. Although the micelles could increase the amount of SCH at the tumor site, the limited efficacy of SCH alone led to difficulty in tumor suppression. Mice treated with free DOX and SCH plus microwave hyperthermia (DS + MW) showed a tumor inhibition rate of about 47.8%. Importantly, drug-loaded micelles plus microwave hyperthermia (DSM + MW) exhibited a better efficacy (~73.5%). It is worth noting that, in comparison with the DSM + MW group, the E-selectin-modified drug-loaded micelles combined with microwave hyperthermia (ES-DSM + MW) group presented a better tumor inhibition effect (about 87.7%), which was due to the satisfactory tumor-targeting efficiency of ES-DSM mediated by leukocytes. In addition, when applied without microwave radiation, ES-DSM treated mice exhibited a poor antitumor effect with an inhibition rate of about 33.6%, which was because the drugs were trapped in the micelles without hyperthermia stimulation and could not be released to execute their function. Further, the E-selectin-modified DOX-loaded micelles supplemented with microwave radiation (ES-DM + MW) group exhibited an ~49.8% tumor inhibition rate, which was not as effective as that of ES-DSM + MW group, suggesting SCH could promote the antitumor efficacy of DOX. Moreover, there was a negligible antitumor effect when CD8[+] T cells of mice were depleted (ES-DSM + MW + anti-CD8), indicating that CD8[+] T cells were indispensable for the antitumor efficacy. Furthermore, the survival time of mice in the ES-DSM + MW group was significantly prolonged compared to that of the other groups (Fig. 6g). Further, tumor tissues of different groups were collected and used for pathological study. TUNEL (Fig. 6j) and H&E (Supplementary Fig. 14) staining of tumor tissues definitely proved that ES-DSM + MW led to a large amount of cell apoptosis and necrosis compared to that in the other groups.

Metastasis is one of the most important reasons for high mortality in cancer patients. Therefore, pulmonary metastasis in each group of mice was evaluated. At the end of the observation period, lung tissues were collected for the observation of metastatic tumor nodules. Figure 6i and k suggested that ES-DSM applied with microwave hyperthermia remarkably suppressed pulmonary metastasis compared to other treatments. This conclusion was further verified by the H&E staining of lung tissues (Fig. 6l). All of these

results indicated that ES-DSM + MW efficiently prevented pulmonary metastasis in tumor-bearing mice.

**Immune response elicited by ES-DSM with microwave radiation (MW).** Further, the in vivo immune response elicited by ES-DSM + MW was investigated. First, mature DCs in tumors and sentinel lymph nodes (SLNs) were analyzed by flow cytometry. As exhibited in Supplementary Figs. 15 and 16, biomarkers of mature DCs (CD80[+] and CD86[+]) in the ES-DSM + MW group were significantly higher than those in the other groups. Since primary CTLs (CD8[+] T cells) responses are important in suppressing tumor growth and helper T cells (CD4[+] T cells) play important roles in the regulation of adaptive immunity, they are considered critical effectors for cancer immunotherapy[43]. Therefore, at the end of the observation period, PBMCs, spleens (Supplementary Fig. 17), and tumors (Fig. 7a and Supplementary Fig. 18a-b) were obtained from each group, and T cells were measured by flow cytometry. In comparison to the other groups, the ratios of CD3[+]CD4[+] and CD3[+]CD8[+] T cells were considerably increased in the ES-DSM + MW group. In contrast, CD4[+]Foxp3[+] T cells, known as regulatory T cells (Tregs), which can hamper effective antitumor immunity, were significantly decreased in the tumor tissue of the ES-DSM + MW treated group (Fig. 7b and Supplementary Fig. 18c). Further, tumor-specific memory T cells (TMEs) were analyzed by detecting the ratio of CD8[+]CD44[+] T cells. A remarkable increase in the percentage of TMEs in both spleens (Fig. 7c, e) and tumors (Fig. 7d, f) was observed, suggesting strong immune surveillance in mice after ES-DSM + MW treatment. Subsequently, antitumor cytokine levels (TNF-α, IFN-γ, and IL-2) in the serum, spleen, and tumor of mice were measured and displayed in Fig. 7g–i. The results suggested that cytokine levels of mice in the ES-DSM + MW group were the highest, indicating the best antitumor immune response. Taken together, the immune response in the ES-DSM + MW group was stronger than that of the DSM + MW and ES-DM + MW groups, which was due to the better tumor-targeting ability mediated by E-selectin and the anti-immunosuppressive effect of SCH. Moreover, when ES-DSM were applied without MW, the immune response in mice was unsatisfactory because the drugs were difficult to be released from the micelles to execute antitumor functions.

The exposure of DAMPs during tumor ICD was an important factor in eliciting antitumor immunity; therefore, level of CRT in tumor tissues after different treatments were examined. As Fig. 7j displayed, ES-DSM + MW treatment induced dramatic increases of CRT exposure in tumor tissues, supporting the remarkable ICD induction ability of this strategy. Tumor-infiltrating CD8[+] T cells (Fig. 7k), CD69[+] T cells (Supplementary Fig. 19a), and perforin (Supplementary Fig. 19b) were also increased after ES-DSM + MW treatment. In contrast, the biomarker of Tregs, Foxp3, was significantly reduced (Fig. 7l). Altogether, these results demonstrated

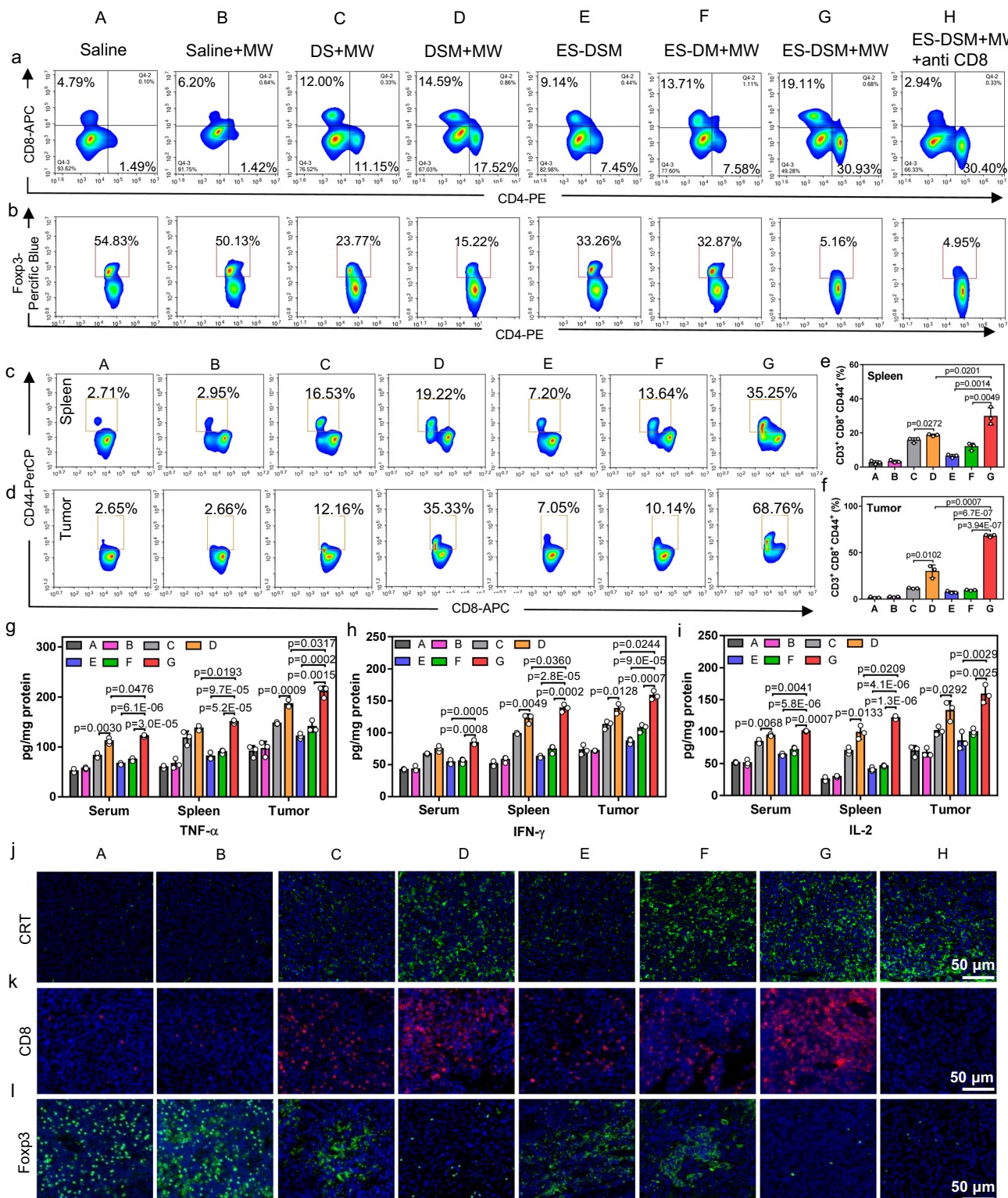

**Fig. 7 Evaluation of the immune response after different treatments in 4T1 tumor models.** The ratios of **a** CD3$^+$CD4$^+$ and CD3$^+$CD8$^+$ T cells in tumors, **b** CD4$^+$ Foxp3$^+$ T cells in tumors, **c** CD8$^+$ CD44$^+$ T cells in spleens, and **d** CD8$^+$ CD44$^+$ T cells in tumors were analyzed by flow cytometry at the end of the observation period. Percentages of CD8$^+$ CD44$^+$ T cells in **e** spleens and **f** tumors were calculated based on (**c**) and (**d**), respectively ($n = 3$). Antitumor cytokine levels, including **g** TNF-α, **h** IFN-γ, and **i** IL-2, in the serum, spleen, and tumor of mice from each group were determined by ELISA assay ($n = 3$). Immunofluorescence was used to examine the levels of **j** CRT, **k** CD8$^+$ T cells, and **l** Foxp3$^+$ T cells in tumor sections at the end of the observation period. DS: free DOX and SCH; DSM: DOX and SCH co-loaded micelles; ES-DSM: E-selectin-modified co-loaded micelles; ES-DM: E-selectin-modified DOX-loaded micelles; MW: microwave radiation (8 W, 30 min). Data are presented as mean values ± SEM and unpaired two-tailed *T* test was performed in (**e**–**i**). The experiments in (**j**), (**k**), and (**l**) showed similar results in three independent mice. Source data are provided as a Source data file.

that the combination of ES-DSM and microwave thermotherapy induced strong ICD and generate a robust immune response at the tumor site. Further, in order to support the importance of ICD during the tumor immunotherapy, the anti-CRT antibody and ecto-ATPase CD39 were intraperitoneally (i.p.) injected every 3 days to block CRT and metabolize ATP starting on day −3. As displayed in Supplementary Fig. 20, the ES-DSM + MW + CD39/anti-CRTα group showed poorer tumor-inhibiting efficacy compared with the group of ES-DSM + MW. Moreover, the infiltrations of CD11c$^+$ DCs, CD3$^+$CD4$^+$, and CD3$^+$CD8$^+$ T cells in tumors of the ES-DSM + MW + CD39/anti-CRTα group were significantly decreased due to the deactivation of ATP and CRT (Supplementary Fig. 21), demonstrating that the ICD of tumor was critical during the tumor immunotherapy.

**Antimetastasis, antirecurrence, and antirechallenge efficacy of ES-DSM with microwave radiation (MW).** To further confirm the treatment efficacy of ES-DSM + MW on the inhibition of pulmonary metastasis, a 4T1 pulmonary metastatic tumor model was established by injecting Luc-4T1 cells into mice via the tail vein, followed by different treatments (Fig. 8a). Pulmonary metastatic tumors of mice in each group were monitored by the bioluminescence signal at days 5, 10, and 20, and the lungs were isolated for bioluminescence imaging at day 20. As displayed in Fig. 8b, c and Supplementary Fig. 22, treatment with ES-DSM + MW showed the strongest antitumor efficacy against pulmonary metastatic tumors. However, the ES-DM + MW group exhibited a poor antimetastatic effect because immunosuppression could not be alleviated and the antitumor immune response cannot be activated effectively in the absence of SCH.

Moreover, a recurrent and rechallenged tumor model was established and treated as shown in Fig. 8d. After different treatments, 90% of the primary tumor was removed surgically on day 12. The residual tumor bed was further monitored and the growth of recurrent tumor was displayed in Fig. 8e, which suggested that ES-DSM + MW treatment significantly inhibited the recurrence of tumor after surgery, followed by the DSM + MW group. Meanwhile, a second 4T1 tumor was inoculated on the other side of mice on day 12 and the growth of the rechallenged tumor was shown in Fig. 8f. Similarly, the growth of the rechallenged tumor in the ES-DSM + MW group was the most inhibited, but treatment with ES-DM + MW did not arrest the growth of rechallenged tumor. The growth of recurrent and rechallenged tumors depended on the level of immune memory after different treatments. As the remarkable increase in the TME percentage was demonstrated in mice treated with ES-DSM + MW (Fig. 7c–f), the residual tumor bed and the second inoculated tumor could be recognized and killed immediately by TMEs. In addition, the infiltrating CD8$^+$ T cells in rechallenged tumor were remarkably increased in the ES-DSM + MW group, while Foxp3$^+$ T cells (Tregs) were greatly reduced (Fig. 8g), further emphasizing the importance of the immune response in the antitumor process. But when the second tumor was inoculated with antigenically different CT26 cells, the rechallenged CT26 tumor could not be effectively suppressed (Supplementary Fig. 23).

**Biocompatibility.** Equally important, the biocompatibility of the various treatments was also verified by hemolysis assay and H&E staining. There was no hemolysis caused by the drug-loaded micelles (Supplementary Fig. 24). In comparison to the cardio-toxicity of free drugs, the major organs of mice in the drug-loaded micelles treated groups appeared to be normal, without obvious histopathological abnormalities, degeneration, or lesions, indicating that no cellular or tissue damage occurred (Supplementary Fig. 25).

## Discussion

In summary, we developed E-selectin-modified thermal-sensitive micelles to co-deliver a chemotherapy agent (DOX) and an immune checkpoint inhibitor (SCH 58261). After intravenous administration, the fabricated ES-DSM can hitchhike with leukocytes mediated by E-selectin to achieve a higher accumulation of drugs at the tumor site. Then, local microwave irradiation can be applied to induce hyperthermia and accelerate the release rate of drugs. Rapidly released DOX can not only directly kill tumor cells but can also improve the immunogenicity of tumors by inducing ICD. Released DAMPs facilitate the maturation and antigen presentation of DCs, further eliciting tumor-specific T-cell immunity. On the other hand, the released SCH can prevent the engagement of ADO with A2AR on the surface of various immune cells, which can liberate the antitumor responses of DCs and CTLs while hampering the activity of Tregs. Consequently, tumor immunosuppression is relieved, and DOX-induced tumor-specific cellular immunity is enhanced. Ultimately, considerably enhanced antitumor efficiency will be achieved via the synergistic effect of chemo-immunotherapy.

Furthermore, due to the maneuverability of drug loading and thermal-sensitive characteristic, the micelles provide more opportunities in the field of drug co-delivery and controlled drug release, while reducing drug leakage during the circulation and avoiding toxic effects. In addition to A2AR antagonists, some other immunotherapeutic drugs, such as antagonists of STING, TLR, PD-1/PD-L1, and other targets, can also be delivered by this kind of smart nano systems, thereby increasing tumor accumulation and decreasing systemic toxicity. The combination of immunotherapy and other therapeutic drugs can also be achieved because the thermal-sensitive micelles allow for the co-delivery of multiple drugs to improve therapeutic efficacy. Overall, the designed micelles for drug co-delivery not only eliminates the paradoxes between ICD-induced antitumor immunity and adenosine-mediated immunosuppression proposed in this article, thus improving antitumor efficacy, but also provides an effective strategy for targeted delivery and spatiotemporally controlled release of other drugs.

## Methods

**Materials.** Acrylonitrile (AN) was purchased from Qinghongfu Technology Co., Ltd. (Beijing, China) and purified by atmospheric distillation before use. Acrylamide (AAm), 4,4′-azobis (4-cyanovaleric acid) (ACVA), dimethyl sulfoxide (DMSO), and azelaic acid were provided by Aladdin (Shanghai, China). The amino polyethylene glycol amine (H$_2$N-PEG-NH$_2$) (Mw = 5 kDa) was purchased from ToYongBio Tech. Inc. (Shanghai, China). Nα,Nα-Bis (carboxymethyl)-L-lysine (NTA) was obtained from Energy Chemical (Shanghai, China). Doxorubicin hydrochloride and indocyanine green (ICG) were brought from Meilun Bio-technology Co., Ltd. (Dalian, China). SCH 58261 was purchased from TCI (Tokyo, Japan). Nile red was obtained from Aladdin (Shanghai, China). Recombinant mouse E-selectin Fc chimera (ES) was from R&D Systems (Minneapolis, USA). 5′-(N-ethylcarboxamido) adenosine (NECA) was bought from ApexBio Technology LLC (Houston, USA). RPMI 1640 medium and fetal bovine serum (FBS) obtained from Sigma (St. Louis, MO, USA) and Sijiqing Biological Engineering Materials Co. Ltd. (Hangzhou, China), respectively. The ELISA kits were all purchased from Meimian Industrial Co., Ltd. (Jiangsu, China). The ATP assay kit was bought from Beyotime (Shanghai, China).

**Cell culture and animals.** The murine 4T1 breast cancer cells (Serial: TCM32) and CT26 colon cancer cells (Serial: TCM37) were obtained from Chinese Academy of Sciences Cell Bank (Shanghai, China), and Luc-4T1 cells (Serial: CM-2233) were purchased from Mingjing Biology (Shanghai, China). Cells were cultured in RPMI 1640 medium supplemented with 10% (v/v) FBS and penicillin/streptomycin (100 U/mL of each) and maintained in the cell incubator (37 °C and 5% CO$_2$). The cells are regularly split using trypsin/EDTA. For the hyperthermia-treated groups, the cells were placed in the cell incubator (43 °C and 5% CO$_2$, 30 min) immediately after adding the test agents, followed by incubation at 37 °C for pre-set time period.

Balb/c mice (female, 6–8 weeks old, 18–20 g) were purchased from Slack Laboratory Animal Co., Ltd (Shanghai, China). Animals were housed at ~22 ± 2 °C, humidity 50 ± 10% on a 12-h light/12-h dark cycle. All animal experiments were performed in accordance with the National Institutes of Health Guide for the Care

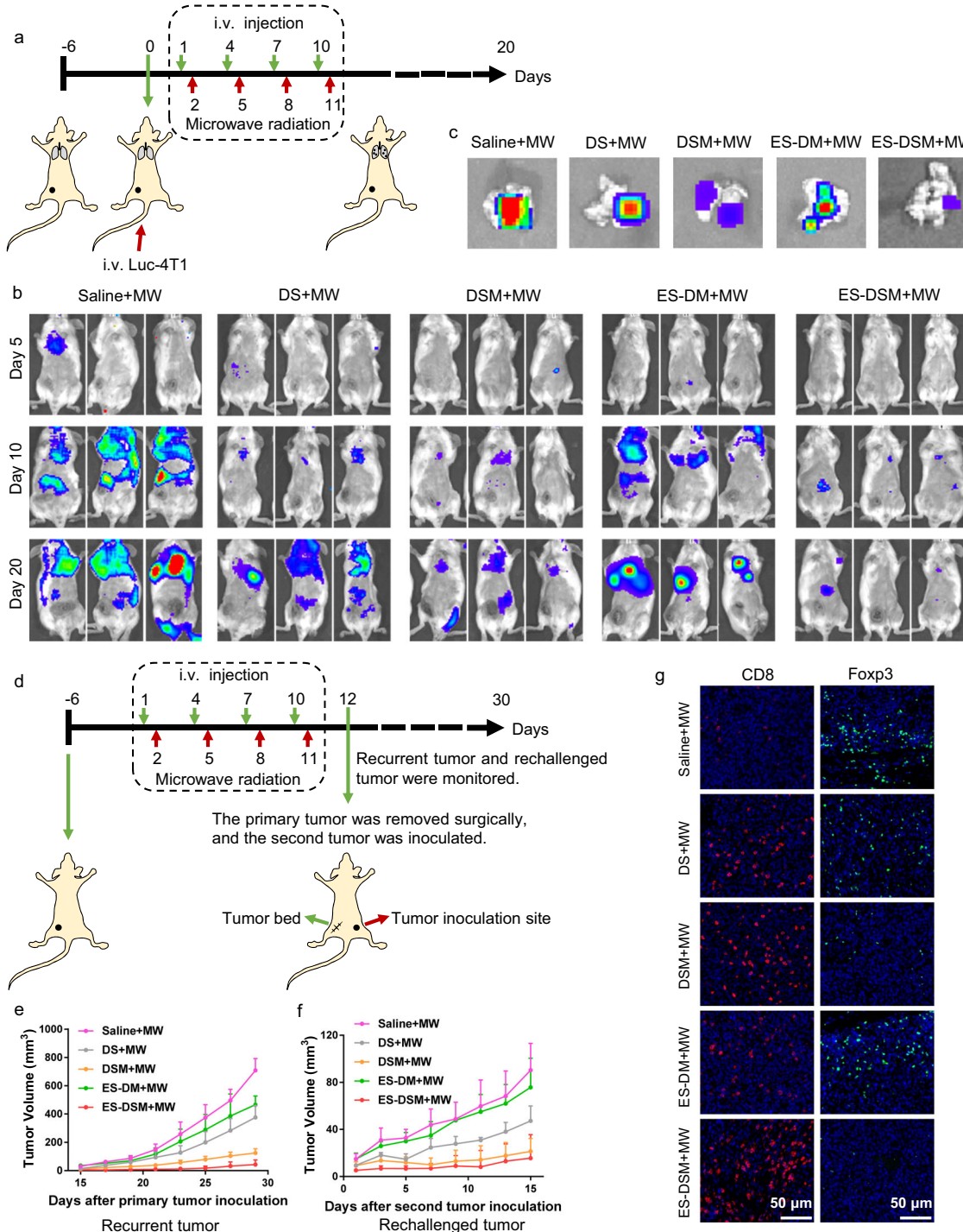

**Fig. 8 Observation of pulmonary metastasis and the growth of recurrent and rechallenged tumors. a** Schematic of the treatment regimen for the pulmonary metastatic model. **b** Luciferase bioluminescence images of Luc-4T1 pulmonary metastatic tumor during the treatments ($n = 3$). **c** Representative luciferase bioluminescence images of lungs on day 20 after different treatments. **d** Schematic of the treatment regimen for the recurrent and rechallenged tumor models. Curves showing volumes of **e** recurrent and **f** rechallenged tumors of mice after various treatments ($n = 5$, data are presented as mean values ± SEM). **g** Immunofluorescence was used to examine CD8+ T cells and Foxp3+ T cells in rechallenged tumor sections at the end of the observation period. The experiments in **g** showed similar results in three independent mice. DS: free DOX and SCH; DSM: DOX and SCH co-loaded micelles; ES-DSM: E-selectin-modified co-loaded micelles; ES-DM: E-selectin-modified DOX-loaded micelles; MW: microwave radiation (8 W, 30 min). Source data are provided as a Source data file.

and Use of Laboratory Animals with the approval of the Scientific Investigation Board of Zhejiang University, Hangzhou, China.

**Synthesis and characterization of NTA-PEG-p-(AAm-co-AN).** First, p-(AAm-co-AN) with a UCST of 43 °C was synthesized by solution copolymerization of AN

and AAm initiated by ACVA. Briefly, 10.95 g (150 mmol) of AAm was weighed into a 500-mL three-necked flask and dissolved in 170 mL of anhydrous DMSO. Subsequently, 2.55 g (50 mmol) of AN was added. Nitrogen was pumped for 1 h to remove the oxygen from the system. After that, 30 mL of separately degassed anhydrous DMSO containing 0.519 g (1.853 mmol) of ACVA was dropped into the system through a constant pressure dropping funnel. Then placed the flask into a

water bath which had been preheated to 65 °C. The reaction mixture was subsequently polymerized for 8 h under nitrogen protection and rapidly cooled to room temperature in an ice bath. The product was precipitated in 10-fold excess volume of methanol. The precipitate was then washed thrice with methanol and dried in a vacuum oven at 70 °C for 24 h.

Next, the $H_2N$-PEG-$NH_2$ was introduced to p-(AAm-co-AN) through the chemical reaction between one of the amine groups in $H_2N$-PEG-$NH_2$ and the carboxyl groups of p-(AAm-co-AN). Briefly, 500 mg (0.1 mmol) of p-(AAm-co-AN) was weighed into a 50-mL flask and dissolved in 10 mL of DMSO, to which 95 mg (0.5 mmol) of EDC and 57 mg (0.5 mmol) of NHS was added and stirred at room temperature for 4 h. Subsequently, the mixture solution was added dropwise to 10 mL DMSO containing 500 mg (0.2 mmol) of $H_2N$-PEG-$NH_2$ (Mw = 5 kDa) at 50 °C. The reaction mixture was stirred for 48 h and then dialysis against deionized water with a dialysis membrane (MWCO: 8–14 kDa) for 48 h, followed by lyophilization and the PEG-p-(AAm-co-AN) was obtained.

Then the NTA was grafted onto PEG-p-(AAm-co-AN) with azelaic acid as the linker. Briefly, 19 mg (100 μmol) of azelaic acid was dissolved in 10 mL of DMSO, to which 20 mg (100 μmol) of EDC and 11.5 mg (100 μmol) of NHS was added and stirred at room temperature for 10 h to activate one of the carboxyl groups of azelaic acid. Subsequently, 500 mg (33.5 μmol) of PEG-p-(AAm-co-AN) was dissolved in 10 mL of DMSO and added dropwise into the above mixture solution, 67 μmol of triethylamine was also supplemented. The reaction mixture was stirred for 17 h at room temperature and then dialysis against deionized water with a dialysis membrane (MWCO: 3.5 kDa) for 48 h, followed by lyophilization to afford the carboxyl-containing PEG-p-(AAm-co-AN). Next, 420 mg (28 μmol) of carboxyl-containing PEG-p-(AAm-co-AN) was dissolved in 10 mL of DMSO, 54 mg (280 μmol) of EDC and 32.5 mg (280 μmol) of NHS was added and stirred at room temperature for 4 h. Then 147 mg (560 μmol) of NTA and 1.12 mmol of triethylamine were dissolved in 10 mL of DMSO/$H_2O$ mixed solution (DMSO: $H_2O$ = 3:2), added dropwise into the above solution and reacted at room temperature for 24 h. After dialysis against deionized water with a dialysis membrane (MWCO: 3.5 kDa) for 48 h and lyophilization, the final product NTA-PEG-p-(AAm-co-AN) was afforded.

The $^1H$-NMR spectra of the polymers were obtained using an NMR spectrometer (AC-80, BrukerBioSpin, Germany) and the spectra were analyzed by MestReNova 6.1.1 software. p-(AAm-co-AN), PEG, PEG-p-(AAm-co-AN) and NTA-PEG-p-(AAm-co-AN) were dissolved in DMSO-$d6$ at concentrations of 20 mg/mL. The molecular weights of p-(AAm-co-AN) and PEG-p-(AAm-co-AN) were analyzed using gel permeation chromatography (GPC) with DMSO as an eluent. PLgel MIXED-C columns (particle size: 5 μm; dimensions: 7.5 mm × 300 mm) that had been calibrated with narrow dextran monodisperse standards were employed with a differential refractive index detector. The flow rate was 0.6 mL/min. Dispersed the polymers in water at a concentration of 2 mg/mL to facilitate the determination of UCST value, the optical transmittance of polymer solutions at different temperatures was measured at 637 nm using an ultraviolet-visible spectrophotometer (UV-2401, Shimadzu, Japan). The UCST value of p-(AAm-co-AN) was determined at the temperature when the optical transmittance became constant. The critical micelle concentration (CMC) of NTA-PEG-p-(AAm-co-AN) was determined using fluorescence spectroscopy and pyrene as a probe. Pyrene was first dissolved in acetone at a concentration of 0.0012 mg/mL and added into tubes. Following evaporation of the acetone at 50 °C, 5 mL of polymer solutions at different concentrations ranging from 2 to 1000 μg/mL were added. After the solution was treated with water bath ultrasonication for 30 min, the emission spectra were recorded on a fluorescence spectrophotometer (F-2500, Hitachi High-Technologies Co., Japan) at room temperature. The excitation wavelength was 336 nm, and the slit widths were set at 10 nm (excitation) and 2.5 nm (emission). The pyrene emission was monitored over a wavelength range of 360–450 nm. From the pyrene emission spectra, the intensity ratio of the first peak ($I_1$, 374 nm) to the third peak ($I_3$, 384 nm) was analyzed and used to calculate the CMC, and the result was calculated by Microsoft Excel 2019.

**Thermal sensitivity of blank micelles**. The NTA-PEG-p-(AAm-co-AN) was dispersed in water at a concentration of 0.5 mg/mL, followed by 30 rounds of probe-type ultrasonic treatment (pulsed every 2 s for a 3 s duration, 400 W). After stirring at 25 °C for 0.5 h, the blank micelles solution was obtained. The blank micelles solution was quartered and incubated at different temperatures (25, 37, 43, 50 °C) for 0.5 h, dropped onto the preheated copper grids, and dry at the corresponding temperature. Subsequently, the morphologies of blank micelles at different temperatures were observed by TEM.

**The ability of micelles to chelate $Ni^{2+}$**. Three milligrams of NTA-PEG-p-(AAm-co-AN) was dispersed in 1 mL of water and treated by probe ultrasound for 30 rounds, stirring at 25 °C for 0.5 h. Then 0.45 mg of $NiCl_2 \cdot H_2O$ was added and the mixture was stirred for another 2 h. After dialyzing against water (MWCO: 3.5 kDa) to remove the excess $Ni^{2+}$ and lyophilizing, the $Ni^{2+}$ content in the micelles was detected by Inductively Coupled Plasma Mass Spectrometry (ICP-MS) (NexION300X, PerkinElmer, USA). The NTA-PEG-p-(AAm-co-AN) without $NiCl_2 \cdot H_2O$ served as a control.

**Preparation and characterization of E-selectin modified DOX/SCH co-loaded micelles (ES-DSM)**. The DOX used in the preparation of drug-loaded micelles was obtained by the reaction between DOX·HCl and two molar equivalents of triethylamine in DMSO for 24 h. Dialysis against water to precipitate the insoluble DOX, followed by centrifuging and lyophilizing to obtain DOX powder for further use. Twenty milligrams of NTA-PEG-p-(AAm-co-AN) was dispersed in 3 mL of water and treated by probe ultrasound for 30 rounds, stirring at 25 °C for 0.5 h to form the stable blank micelles. DOX and SCH 58261 (SCH) were dissolved together in DMSO at the final concentrations of 0.8 and 0.2 mg/mL, respectively. Then 1 mL of DMSO solution of DOX/SCH was added dropwise to micelles solution with constant stirring (DOX:SCH:polymer = 4:1:100). Subsequently, 3 mg of $NiCl_2 \cdot H_2O$ was added and the mixture was stirred at 25 °C for another 2 h, followed by dialyzing against water (MWCO: 3.5 kDa) for 24 h and centrifuging at 1800 × $g$ for 10 min to eliminate aggregates of non-encapsulated DOX/SCH. Ultimately, the solution of DOX/SCH co-loaded micelles (DSM) was lyophilized and stored at 4 °C. E-selectin could be introduced onto the surface of DSM between the interaction of His-tag of E-selectin and Ni-NTA of polymer. Briefly, different concentrations of E-selectin (0, 0.1, 0.2, 0.5, 1, 2, 3 μg/mL) were added to the DSM solution (at a polymer concentration of 1 mg/mL), respectively, incubated at 37 °C for 1 h and further in 4 °C overnight to afford the E-selectin modified DSM (ES-DSM). The preparation of DOX-loaded micelles (DM and ES-DM) were the same as above, except the absence of SCH. The particle sizes and zeta potentials of DSM and ES-DSM were recorded by dynamic light scattering (DLS) (Zetasizer, 3000HS, 66 Malvern Instruments Ltd.). The morphology of ES-DSM was observed by transmission electron microscopy (TEM) (JEOL JEM-1230, Japan). The encapsulation efficiency (EE) and drug loading (DL) were determined by fluorospectro photometer (DOX: Ex = 480 nm, Em = 560 nm, Slit width = 5 nm; SCH: Ex = 320 nm, Em = 385 nm, Slit width = 5 nm). Briefly, the drug-loaded micelles were disrupted by DMSO and the total DOX and SCH contents were quantified. EE% and DL% were calculated by the following formulas:

$$EE\% = \frac{\text{mass of drug encapsulated into micelles}}{\text{mass of drug added}} \times 100\% \quad (1)$$

$$DL\% = \frac{\text{mass of drug encapsulated into micelles}}{\text{mass of drug-loaded micelles}} \times 100\% \quad (2)$$

**Thermal-triggered size changes of micelles**. The size changes of micelles in response to temperature were monitored by DLS. The sizes of blank micelles, DSM and ES-DSM in different temperatures (5, 15, 25, 37, 43, 50 °C) were measured. The samples (at a polymer concentration of 1 mg/mL) were incubated at the corresponding temperature for 10 min before measurement. There are three repeat groups for each sample.

**Thermal-sensitive in vitro drug release behavior of ES-DSM**. The DOX and SCH release profiles of ES-DSM in different temperatures were tested by dialysis method. The dialysis bags (MWCO: 3.5 kDa) containing 1 mL of free DOX and SCH (DS), and ES-DSM (concentrations of DOX and SCH were 90 and 15 μg/mL, respectively) were immersed into falcon tubes containing 30 mL PBS (pH 7.4). These tubes were put into incubator shakers (37 and 43 °C, respectively) and horizontally shaken at 60 rpm/min. At each pre-set time point, the release media were collected and replaced with fresh PBS. The DOX and SCH contents in the release media were detected by fluorospectro photometer. Each time point was performed trice.

**Biocompatibility of micelles on leukocytes**. To obtain the leukocytes, the blood of mice was taken by excising eyeballs and the leukocytes were isolated by the mouse peripheral blood leukocyte separation kit according to the manufacturer's instructions (Solarbio, China). The cytotoxicity of blank micelles to leukocytes was measured by CCK-8 assay. Briefly, leukocytes were suspended in RPMI 1640 medium and seeded in 96-well plate at a density of $1 \times 10^4$ cells per well and then exposed to blank micelles at a series of concentrations (0, 100, 200, 400, 800, 1000 μg/mL) for different time (1, 2, 4, 8 h). Subsequently, 10 μL of CCK-8 solution was added and incubated for 1 h, followed by measuring the absorbance of each well by microplate reader at 450 nm. Cell viability was calculated in reference to negative cells without exposure to test agents. All experiments were repeated thrice.

The cytotoxicity of ES-DSM to leukocytes was then tested. The leukocytes were seeded in 96-well plate at a density of $1 \times 10^4$ cells per well and exposed to ES-DSM at different DOX concentrations (6.5, 12.5, 25, 31.5, 37.5 μg/mL) for various time (1, 2, 4, 8 h). The cell viabilities were measured by CCK-8 assay. All experiments were repeated thrice.

The chemotaxis and penetration ability of leukocytes were investigated by transwell migration assay (pore size of transwell polycarbonate membrane was 8 μm). Briefly, leukocytes were exposed to DSM or ES-DSM at a DOX concentration of 37.5 μg/mL for 8 h. After washing thrice by PBS, the leukocytes were suspended in RPMI 1640 medium added to the upper chamber of transwell which had been attached by HUVECs. The lower chamber of transwell was filled with RPMI 1640 medium containing chemokines (10 ng/mL of CXCL2 and 100 ng/mL of CXCL12). After 4 h of incubation, the leukocytes in the lower chamber were observed by

microscopy, then the leukocytes were collected and the numbers were counted by cytometry. The transwell percentage was calculated by the formula:

$$\text{Transwell}\% = \frac{N_{\text{lower champer}}}{N_{\text{total}}} \times 100\% \qquad (3)$$

Simultaneously, mice were intravenously injected with DSM or ES-DSM and the leukocytes were isolated after 24 h. The chemotaxis and penetration ability of the isolated leukocytes were also analyzed by transwell as mentioned above, and the transwell percentage was calculated too.

**Leukocyte-adhering ability of ES-DSM.** Two hundred microliters of DSM or ES-DSM (concentrations of DOX and SCH were 300 and 50 µg/mL, respectively) was injected into the mice via the tail vein, and at 2, 8, and 24 h after injection, the leukocytes of treated mice were isolated by the mouse peripheral blood leukocyte separation kit according to the manufacturer's instructions (Solarbio, China). The DOX fluorescence on the obtained leukocytes was analyzed by flow cytometry (ACEA NovoCyte, USA) and confocal laser scanning microscope (CLSM) (Leica SP8, Germany). In addition, leukocytes were isolated 24 h after ES-DSM injection, the T lymphocytes and neutrophils were labeled by APC-anti-CD3 (Cat#100235, 1:40) or CD16 (Cat#158005, 1:40) antibody (BioLegend, USA), respectively, then observed by CLSM.

**Thermal-sensitive drug release behavior of micelles at cellular level.** First, Nile red was loaded into the micelles. The preparation of Nile red-loaded micelles was the same as DSM, excepted the model drug used was Nile red instead of DOX/ SCH. 4T1 cells were suspended in RPMI 1640 medium and seeded in 12-well plate at a density of $1 \times 10^5$ cells per well and allowed to attach overnight. Subsequently, the cells were treated with free Nile red or Nile red-loaded micelles (at a final Nile red concentration of 0.1 µg/mL) and the hyperthermia-treated groups were placed in the cell incubator (43 °C and 5% $CO_2$, 30 min) immediately, followed by incubation at 37 °C for 6 h. After washed trice with PBS, the cells were harvested and fluorescence intensity was detected by flow cytometry. Besides, the cell fluorescence was also observed by CLSM. After incubation and washed twice with PBS, the cells were fixed and the nuclei were stained by DAPI, followed by CLSM observation.

Then, DOX was loaded into the micelles. The free DOX and DOX-loaded micelles were added to 4T1 cells at a final DOX concentration of 4.5 µg/mL. After treated with hyperthermia and 6 h incubation, the cells were washed trice with PBS and fixed. After staining by DAPI, the cells were observed by CLSM.

**Cytotoxicity and apoptosis.** First, the cytotoxicity of blank micelles was measured by MTT assay. 4T1 cells were suspended in RPMI 1640 medium and seeded in 96-well plate at a density of $1 \times 10^4$ cells per well and allowed to attach overnight. Then the cells were exposed to blank micelles at a series of concentrations (0, 100, 200, 400, 600, 800, 1000 µg/mL) for 48 h. The hyperthermia-treated groups were placed in the 43 °C cell incubator for 30 min, followed by incubation at 37 °C until 48 h. Subsequently, 20 µL of 3-(4,5-Dimethyl-2-thiazolyl)-2,5-diphenyl-2H-tetra-zolium bromide (MTT) solution (5 mg/mL) was added to each well for an additional 4-h incubation at 37 °C. After that, the medium was replaced with 100 µL of DMSO to dissolve the purple formazan crystals in the bottom of the well. The plate was shaken for 30 min, and the absorbance of the solution in each well was measured by microplate reader at 570 nm. Cell viability was calculated in reference to negative cells without exposure to test agents. All experiments were repeated thrice.

Subsequently, the cytotoxicity of free DOX/SCH (DS), DSM, and ES-DSM combined with or without hyperthermia were determined by MTT assay. 4T1 cells were suspended in RPMI 1640 medium and seeded in 96-well plate at a density of $1 \times 10^4$ cells per well and allowed to attach overnight. Then the cells were exposed to DS, DSM, or ES-DSM at different drug concentrations for 48 h (the concentration ratio of DOX and SCH is 6:1). The hyperthermia-treated groups were immediately placed in the cell incubator which had pre-set to 43 °C for 30 min after exposing to the test agents, followed by incubation at 37 °C until 48 h. Cell viability was measured as described above.

Cell apoptosis induced by DS, DSM, and ES-DSM combined with or without hyperthermia was investigated by flow cytometry. 4T1 cells were suspended in RPMI 1640 medium and seeded in 12-well plate at a density of $1 \times 10^5$ cells per well and allowed to attach overnight. Subsequently, the cells were exposed to DS, DSM, or ES-DSM (concentrations of DOX and SCH were 4.5 and 0.75 µg/mL, respectively) and treated with or without hyperthermia. After a 24-h incubation, cells were harvested and stained by the Annexin V-FITC/PI apoptosis detection kit (Beyotime Biotech, China) according to the manufacturer's instructions, followed by flow cytometer analysis.

**Detection of the ICD biomarkers.** The exposure of DAMPs (CRT, HMGB1, and ATP) of tumor cells after different treatment were detected. Briefly, 4T1 cells were treated with DS, DSM, or ES-DSM (concentrations of DOX and SCH were 4.5 µg/mL and 0.75 µg/mL) with or without hyperthermia. The exposure of CRT was observed by the immunofluorescence via CLSM at the time of 12 h (Calreticulin Rabbit Monoclonal Antibody, Cat#AF1666, 1:500, Beyotime, China). Semi-quantitative analysis was performed using Image J software. After

incubating for 48 h, the cell culture supernatant was collected, the content of ATP was detected by ATP assay kit and HMGB1 was detected by ELISA kit, according to manufacturer's instructions.

**Co-incubation of tumor cells and bone marrow-derived DCs.** The murine bone marrow-derived DCs (BMDCs) were isolated from 6-week old Balb/c female mice. Briefly, the bone marrow of mice was collected via flushing the femurs and tibias with PBS, and red blood cells were lysed. The remaining cells were washed twice with PBS and cultured in the complete RPMI 1640 medium containing recombinant murine GM-CSF (20 ng/mL) (MedChemExpress, USA) for 6 days to acquire the immature DCs. On day 7, the immature DCs were co-incubated with 4T1 cells which had been previously treated with PBS, DS, DSM, or ES-DSM (concentrations of DOX and SCH were 4.5 µg/mL and 0.75 µg/mL, respectively) (supplemented with or without hyperthermia) 24 h ago. After a 48-h co-incubation, DCs were stained with the indicated antibodies including PE-CD80 (Cat#104707, 1:40, Bio-Legend, USA), APC-CD86 (Cat#105011, 1:40, BioLegend, USA), and PE-MHC II (Cat#12-5321-81, 1:1000, ThermoFisher, USA), analyzed by flow cytometry. In addition, the cytokine levels in the supernatant of the co-incubation system including IL-12p70, IL-6, and IL-10 were detected using ELISA kits according to manufacturer's instructions.

Besides, the immature DCs were co-incubated with 4T1 cells which had been previously treated with D (DOX alone), DS, DM, DSM, ES-DM, or ES-DSM (concentrations of DOX and SCH were 4.5 µg/mL and 0.75 µg/mL, respectively) and supplemented with hyperthermia 24 h ago. After a 48-h co-incubation with the presence of 1 µM (a dose that mimics the concentration of adenosine found in the tumor microenvironment) of NECA (adenosine analog), DCs were stained with the indicated antibodies including PE-CD80 (Cat#104707, 1:40, BioLegend, USA), APC-CD86 (Cat#105011, 1:40, BioLegend, USA), and PE-MHC II (Cat#12-5321-81, 1:1000, ThermoFisher, USA), analyzed by flow cytometry. In addition, the cytokine levels in the supernatant of the co-incubation system including IL-12p70, IL-6, and IL-10 were detected using ELISA kits according to manufacturer's instructions.

**Co-incubation of tumor cells, bone marrow-derived DCs, and spleen lymphocytes.** Spleen lymphocytes were extracted from the spleens of Balb/c mice using lymphocyte density gradient centrifugation with Ficoll-paque PREMIUM. The immature DCs and lymphocytes were co-incubated with 4T1 cells which had been previously treated with PBS, DS, DSM, or ES-DSM (concentrations of DOX and SCH were 4.5 and 0.75 µg/mL, respectively) (supplemented with or without hyperthermia) 24 h ago. After a 48-h co-incubation, lymphocytes were stained with the indicated antibodies including FITC-CD3 (Cat#100204, 1:200), APC-CD8 (Cat#100712, 1:100), PE-CD4 (Cat#100408, 1:100) and Percific Blue-Foxp3 (Cat#126410, 1:50) (BioLegend, USA), analyzed by flow cytometry. In addition, the cytokine levels in the supernatant of the co-incubation system including TNF-α, IL-2, and IFN-γ were detected using ELISA kits according to manufacturer's instructions.

Besides, the immature DCs and lymphocytes were co-incubated with 4T1 cells which had been previously treated with D, DS, DM, DSM, ES-DM, or ES-DSM (concentrations of DOX and SCH were 4.5 and 0.75 µg/mL, respectively) and supplemented with hyperthermia 24 h ago. After a 48-h co-incubation with the presence of 1 µM of NECA, lymphocytes were stained with the indicated antibodies including FITC-CD3 (Cat#100204, 1:200), APC-CD8 (Cat#100712, 1:100), PE-CD4 (Cat#100408, 1:100), and Percific Blue-Foxp3 (Cat#126410, 1:50) (BioLegend, USA), analyzed by flow cytometry. In addition, the cytokine levels in the supernatant of the co-incubation system including TNF-α, IL-2, and IFN-γ were detected using ELISA kits according to manufacturer's instructions.

**Biodistribution of DSM and ES-DSM.** The orthotopic tumor models were established by subcutaneous injection of 4T1 cells ($5 \times 10^5$) dispersed in serum-free RPMI 1640 medium into the third breast pad of Balb/c mice. Treatment began when the tumor volume reached 500 mm³. For the observation and imaging of the micelles biodistribution, ICG-loaded micelles were prepared the same as DSM, and the modification of E-selectin was the same as ES-DSM. Two hundred microliters of ICG-loaded micelles or ES-ICG-loaded micelles were injected into the mice via the tail vein and at 2, 6, 12, 24 h after injection, the treated mice were anesthetized and the fluorescence images were acquired by Maestro in vivo imaging system. 24 h after injection, the mice were sacrificed to harvest the main organs (heart, liver, spleen, lung, kidneys, and tumor). Fluorescence images were acquired, and the fluorescence intensity of these organs was measured ex vivo using an in vivo imaging system. The fluorescence of ICG and CD45 in tumors was analyzed by immunofluorescence and observed by CLSM.

**In vivo antitumor study.** In total, $5 \times 10^5$ of 4T1 cells were orthotopically injected into one of the breast pads of Balb/c mice. After 1 week, the mice were randomly sorted into eight groups (6 mice per group) to, respectively, receive one of the following treatments once every 3 days: Saline, Saline+MW, DS + MW, DSM + MW, ES-DSM, ES-DM + MW, ES-DSM + MW, ES-DSM + MW + anti-CD8, for four times of treatment. 3 mg/kg DOX and 0.5 mg/kg SCH per dose was used in the treatment and at 24 h post-i.v. injection of the test agents, the mild microwave

(MW) was applied locally for 30 min (8 W). The microwave probe was positioned 1 cm away from the fixed animal and oriented toward the orthotopic breast tumor. The anti-CD8 antibody (Cat#BE0004-1, BioXcell, USA) was intraperitoneal (i.p.) injected to deplete the CD8$^+$ T cells on the days of −3 and treated every 3 days until the end of monitoring (100 μg/mice per injection). The body weight and tumor volume were monitored every 2 days and the survival time was monitored. The tumor volume was calculated using the formula: $a^2 \times b/2$, in which a and b represent the smallest and largest diameters of the corresponding tumor, respectively.

In order to assess the efficacy of SCH, 4T1 tumor-bearing mice were randomly sorted into three groups (6 mice per group) to, respectively, receive one of the following treatments once every 3 days: Saline+MW, SCH + MW, ES-SM + MW, for four times of treatment. 0.5 mg/kg SCH per dose was used in the treatment and at 24 h post-i.v. injection of the test agents, the microwave (MW) was applied locally for 30 min (8 W). The tumor volume was monitored every 2 days.

At the end of monitoring on day 23, the mice were sacrificed and main organs (heart, liver, spleen, lung, kidney, and tumor) were harvested and fixed in 4% paraformaldehyde, embedded in paraffin, cut into 5-μm slices, and stained with H&E, then examined under a light microscope. The apoptosis of tumor tissue also be studied by immunofluorescence of TUNEL staining. To demonstrate the ICD of tumor tissues, the CRT exposure level was studied by immunofluorescence and observed by CLSM. To examine the immune response, the infiltration of CD8$^+$ T cells and Tregs (Foxp3) in tumors were analyzed by immunofluorescence, while the infiltration of active T cells (CD69) and perforin were studied by immunohistochemistry. T cells (CD3$^+$CD8$^+$ and CD3$^+$CD4$^+$) in PBMC, spleen, and tumor were isolated by density gradient centrifugation and stained with corresponding fluorescence-labeled antibodies, then analyzed by flow cytometry. The CD3$^+$CD4$^+$Foxp3$^+$ T cells in tumor and CD3$^+$CD8$^+$CD44$^+$ T cells in spleen and tumor were stained with corresponding antibodies mentioned above and evaluated by flow cytometry. Particularly, the CD44 was labeled by PerCP-CD44 Antibody (Cat#103035, 1: 100, BioLegend, USA). Besides, the spleen and tumor were ground in ice bath by homogenizer, the supernatant after centrifugation (16,000 × g, 10 min, 4 ℃) was collected for measurement. Levels of TNF-α, IFN-γ, and IL-2 in serum, spleen, and tumor were examined using the ELISA kits according to manufacturer's instructions. DCs (CD11c$^+$CD80$^+$ and CD11c$^+$CD86$^+$) isolated from tumor and sentinel lymph node (SLN) were also stained with corresponding antibodies and analyzed by flow cytometry.

In a separate experiment to investigate the contribution of ICD to the tumor immunotherapy, 4T1 tumor-bearing mice were randomly sorted into three groups (6 mice per group) to, respectively, receive one of the following treatments once every 3 days: Saline+MW, ES-DSM + MW, ES-DSM + MW + CD39/anti-CRTα, for four times of treatment. The anti-CRT antibody (Cat#ab223614, Abcam, USA) (10 μg/mice per injection) and ecto-ATPase CD39 (Cat#4398-EN-010, R&D System, USA) (1 μg/mice per injection) were intraperitoneally (i.p.) injected every 3 days to block CRT and metabolize ATP starting on day −3. The tumor volume was monitored every 2 days. Mice were sacrificed on day 23, the DCs (CD11c$^+$) and T cells (CD3$^+$CD8$^+$ and CD3$^+$CD4$^+$) in tumors were stained with corresponding antibodies and analyzed by flow cytometry. Besides, the exposure of CRT and infiltration of CD8$^+$ T cells in tumors were analyzed by immunofluorescence.

A lung metastatic model of breast cancer was also established to further investigate the treatment efficacy on metastatic cancer. Initially, the orthotopic breast tumor-bearing mice were established by injecting $5 \times 10^5$ of 4T1 cells. Six days later, $1 \times 10^5$ of Luc-4T1 cells were injected intravenously. Then, the mice were randomly sorted into five groups (3 mice per group) to, respectively, receive one of the following treatments once every 3 days: Saline+MW, DS + MW, DSM + MW, ES-DM + MW, ES-DSM + MW, for four times of treatment. 3 mg/kg DOX and 0.5 mg/kg SCH per dose was used in the treatment and the MW (8 W, 30 min) was applied at 24 h post-i.v. injection of the test agent. The microwave probe was positioned 1 cm away from the fixed animal and oriented toward the orthotopic breast tumor. The growth of pulmonary metastasis tumors was monitored by IVIS Spectrum imaging system (PerkinElmer, USA) after intraperitoneal injection of D-luciferin (15 mg/mL, 200 μL). At the end of monitoring on day 20, the mice were sacrificed and the fluorescence images of lungs were acquired.

Tumor recurrence and rechallenge study were further invested. The orthotopic breast tumor-bearing mice were established as mentioned above and received different treatments. After 4 times of treatment, 90% of the primary tumor was removed surgically on day 12, and the tumor bed was further monitored and the volume of recurrence tumor was calculated every 2 days. Simultaneously, $5 \times 10^5$ of 4T1 cells were inoculated into the breast pads on the other side of mice on day 12. The rechallenged tumor was also monitored every 2 days. At the end of monitoring on day 30, the mice were sacrificed and rechallenged tumor was collected to analyze the infiltration of CD8$^+$ T cells and Tregs (Foxp3) by immunofluorescence. In addition, after orthotopic breast tumor-bearing mice were treated with ES-DSM + MW, $5 \times 10^5$ of CT26 cells were inoculated subcutaneously in the left hind limb on day 12. The rechallenged CT26 tumor was also monitored every 2 days.

**Hemolysis assay.** Fresh mice blood samples stabilized by ethylenediaminetetraacetic acid were obtained and then RBCs were isolated from serum by centrifugation at 250 × g for 15 min. After being washed five times with saline, the purified blood was diluted to a 2% RBC suspension, and then 0.5 mL of the RBC suspension was added to 1.5 mL Eppendorf tubes and mixed with the following

agents: (1) 0.5 mL of saline as a negative control, (2) 0.5 mL of pure water as a positive control, (3) 0.5 mL of blank micelles (M) at 2 mg/mL, (4) 0.5 mL of E-selectin modified blank micelles (ES-M) at 2 mg/mL, (5) 0.5 mL of DSM at a micelle concentration of 2 mg/mL, and (6) 0.5 mL of ES-DSM at a micelle concentration of 2 mg/mL. All the mixtures were vortexed and kept at room temperature for 3 h. Finally, the mixtures were centrifuged at 7200 × g for 5 min and the absorbance of the supernatants was determined at 541 nm using an ultraviolet spectrophotometer. The percent hemolysis of RBCs was calculated as follows:

$$\text{Percent hemolysis } \% = \frac{A_{\text{sample}} - A_{\text{negative}}}{A_{\text{positive}} - A_{\text{negative}}} \times 100\% \quad (4)$$

**Statistical analysis.** Statistical calculations were performed using Prism 7 software (GraphPad). Data were expressed as the mean and SEM. Differences were statistically evaluated by unpaired two-tailed $T$ test. The differences were considered to be statistically significant for a $p$ value of <0.05. To analyze the survival time of mice, Kaplan–Meier survival curves were generated, and Log-rank Mantel–Cox tests were performed. P values of <0.05 were considered significant.

**Reporting summary.** Further information on research design is available in the Nature Research Reporting Summary linked to this article.

## Data availability

All data associated with this study are available within the Article, Supplementary Information, or Source data file. Source data are provided with this paper.

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

## Acknowledgements

This work was supported by National Key Research and Development Projects Inter-governmental Cooperation in Science and Technology of China [2018YFE0126900] and Natural Science Foundation of Zhejiang Province (LD21H300002).

## Author contributions

Y.Z.D. designed and guided the overall research project. J.Q. as the first author designed the experiments and wrote the manuscript. F.Y.J. and Y.C.Y. performed the actuation experiments. Y.D. and D.L. involved the synthesis of polymers and cellular experiments. X.L.X. and J.W. assisted with animal maintenance. L.W.Z., M.J.C., and G.F.S. involved the data analysis and other property characterizations. L.M.W. and J.S.J. provided intellectual input and helped interpret the results.

## Competing interests

The authors declare no competing interests.
