## [Peer Review File · Nature Communications]

Reviewers' Comments:

Reviewer #1:

Remarks to the Author:

The manuscript Synergistic effect of tumor chemo-immunotherapy induced by leukocyte-hitchhiking thermal-sensitive micelles by Qi et al shows that E-selectin-modified thermal-sensitive micelles (ES-DSM) co-delivered doxorubicin and SCH 58261 activated by local microwave irradiation-induced hyperthermia and caused T cell-dependent anticancer immunity and tumor reduction.

The authors show that ES-DSM together with hyperthermia cause immunogenic cancer cell death, dendritic cell maturation and T cell activation in vitro. In vivo ES-DSM caused 4T1 lung tumor growth reduction of pulmonary metastasis which was accompanied by an increase in the CTL over Treg ratio. Finally the authors demonstrate generation of immunological memory upon treatment with ES-DSM and hyperthermia.

The manuscript is well written and the methodology is technically sound, nevertheless the manuscript would benefit from additional linguistic editing and more experimental details in both the material and methods section as well as the figure legends.

The authors should explain limited viability of negative controls in Figure 2 e.

The DAMP release is not well explored.

In vitro the authors should investigate calreticulin (CALR) exposure (not the expression). ATP is commonly measured by luciferase conversion assays yet experimental details on how the authors conducted this assay (ELISA) are missing.

In vivo the authors show an increase in the expression of both HMGB1 and CALR whereas a release of HMGB1 and an exposure of CALR would be expected.

As to show the importance of ICD the authors could use blocking antibody against CALR and ectoATPases against ATP in vivo and analyse their effect on tumor growth.

Yet another convincing assay to demonstrate immunological memory might be the eradication of subcutaneously implanted tumors and rechallenge with antigenetically identical and different syngeneic cancer cells.

Reviewer #2:

Remarks to the Author:

The current manuscript by Qi et al. describes a novel technological advance for the localized delivery of chemo- and immunotherapeutic drugs. Whilst liposome like structures have been developed for the purpose of chemotherapy and SCH58261 before (e.g. PMID 29720380) the novelty of the current approach lies in the nature of its thermal sensitive allowing for its spatial and temporal control. Overall the study is well conducted and introduces novelty to the field but I recommend the following to further improve its significance.

1) Throughout the manuscript in the majority of figures, information is missing on the number of times experiments were performed and how many technical or biological replicates are being presented. This makes it very difficult to assess the robustness of the observations. In addition statistical analysis appears absent from many figures where conclusions have been made based upon the observed differences e.g. Figure 3F-O, 4F-M. This should be remedied by the inclusion of statistical tests and/or additional replicates to ensure statistically meaningful data is presented.

2) Further example is Figure 7B, it is unclear if this experiment was performed on n= 3 mice only, is there a quantification of this particular experiment?

3) Whilst the data conclusively show that DM is superior therapeutically to free dox and DSM is superior to DM the relative therapeutic activity of encapsulated SCH vs free SCH is not formally shown. Whilst this is likely given the pharmacokinetic profile of SCH it would be important to

confirm this given it is a central message for the manuscript

4) In Figure 2e it appears that the DSM and ES-DSM formulations induce apoptosis in the absence of hyperthermia but this does not appear to be reflected in 1f, the quantification of this data. Can this be reconciled?

5) For the in vivo analysis although showing the increased proportion of CD8 and CD4 T cells is suggestive of more T cell infiltrate, it would be more compelling to quantify the number of cells.

6) The paper is quite heavy with acronyms that make it very hard to follow the data in the figures. I would recommend that a key for the formulations is included in each figure (or figure legend) to make this easier to follow.

7) The Conclusion/ Discussion section is very short and misses an opportunity to place this work in the context of the wider field. Some suggested topics to include in this section is how this approach differs from other encapsulation approaches and other immunotherapies that could benefit from such a localised delivery. For example one application would be agonists of innate immunity STING, TLR agonists etc that have toxicity when given systemically.

Reviewer #3:

Remarks to the Author:

Qi et al. present a massive study that examines a combination photo/chemo/nano/immunotherapy for metastatic cancer. Superficially, the final results appear promising, as significant decreases in tumor volume are reported. But the study is so large that none of the results are described in sufficient detail to be reproduced by others, possibly due to space constraints. Critically, the materials are not sufficiently characterized. The key component of the overall strategy is to attach nanoparticles to leukocytes via their known interactions with E selectin, with the hope that nanoparticles containing doxorubicin and a photosensitizer can be transported, i.e. "hitchhike", to tumors via tumor infiltrating leukocytes. But the strength of this attachment is not examined, and the methods employed are not logical. For example, intravenous injection is employed, which would require extremely high concentrations of nanoparticles to sufficiently tag enough leukocytes to reach the small percentage that would actually migrate to tumors. Additionally, it is not clear how the locally applied microwave radiation could result in enhanced killing of metastatic sites. Of note, the lung tumor model employed does not explain how the microwave radiation is applied to the animals and why this treatment locally effects sites of metastasis and does not cause release of the micelles throughout the mouse. Of note, the micelles employed have a high CMC and are likely not sufficiently stable to remain attached to leukocytes as they travel, potentially for days, in vivo before reaching tumors. Importantly, this strategy is not novel and may be fundamentally flawed. Their strategy stems from the seminal work of the Irvine group at MIT that developed nanoparticle "backpacking" technology over a decade ago (DOI: 10.1038/nm.2198), where nanoparticles were carefully attached to T cells for enhanced targeting of tumors. In this previous study, the attachment and duration of attachment were both thoroughly investigated and the T cells were strategically stimulated to home to tumors and sites of metastasis. But in the current manuscript by Qi et al., the use of nanoparticles to attach to E selectin may be fundamentally flawed, as this method may actually serve to block the receptors necessary for the leukocytes to extravasate into tumors. Such a strategy needs much more extensive investigation, yet in the current study, it is not even clear which specific cells are being targeted, as there are numerous immune cells (monocytes, macrophages, DCs, neutrophils, eosinophils, certain T cell subsets... etc) expressing receptors that would bind the E selectin modified nanoparticles, yet these cells and their markers are not reported. For these reasons, rejection of this manuscript is recommended. Further details of some key issues that should be addressed prior to publication are outlined below:

1) Materials are lacking characterization and justification. Why was this specific polymer used, when there are a wide range of thermosensitive materials available?

2) The CMC of the micelles reported is relatively high, suggesting that the particles will not be stable in vivo, particularly for the time required for cells to migrate to tumors. Of note, the seminal

work by the Irvine group required specially crosslinked liposomes for enhanced stability in order to traffic in vivo.

3) The method of attaching E selecting to the nanoparticles is not covalent and appears to rely on a His tag interaction that would not be sufficiently stable in vivo.

4) The fundamental strategy may be flawed: Wouldn't this "hitchhiking" method serve to block receptors necessary for leukocytes to enter inflamed tissues?

5) The stability/kinetics of nanoparticle binding to cell surfaces is not examined or reported. How long do the particles reside on the cell surface? Does receptor turnover result in in endocytosis of the micelles over time?

6) Prior backpacking strategies noted that the payload would release from the attached nanoparticles and locally modulate the cells to which they were attached. In fact, this was a key part of the Irvine strategy, which was to locally modulate T cells with cytokines while attached to them (DOI: 10.1038/nm.2198). In this current work, wouldn't locally attaching micelles containing a cytotoxic agent serve to decrease viability of the leukocytes to which they are attached?

7) The use of an intravenous injection is likely not a viable strategy, as a massive amount of material will have to be injected in order to sufficiently coat enough leukocytes, the vast majority of which will not go to tumors.

8) The confocal images in figure, 1i show extensive labeling of leukocytes. But it is not clear what cell type is being shown, nor how these cells were extracted. Methods simply state that leukocytes were extracted, and do not report how now where they were extracted from. This level of labeling is unexpected following an intravenous injection, and it appears that this may have actually been an in vitro experiment and not in vivo as reported.

9) In general, targeted cells are not sufficiently examined for their activity nor type and few markers are reported throughout the manuscript.

10) The lung metastasis model is lacking in detail. It is not clear how the microwave radiation was applied. Was a single tumor targeted? How were metastatic niches targeted or prevented?

11) Overall, this manuscript appears to be attempting to increase the impact of the work by combining 3-4 separate manuscripts into 1. This work should be broken down into multiple manuscripts so that each step can be more thoroughly examined and reported. As shown, this manuscript lacks detail required to reproduce the reported results in another lab.

12) This work also lacks novelty, as this "hitchhiking" strategy is highly dependent on the work of the Irvine group's nanoparticle "backpacking" technology. Yet the extensive work by the Irvine group as well as several other labs (some of whom have attempted in vivo instead of ex vivo cell tagging) that have adopted this backpacking strategy have not been cited in this manuscript. Authors should thoroughly examine and summarize this prior work, which would provide insight into their strategy and the necessary characterization methods that should be employed for validation.

Reviewers' comments:

Reviewer #1 (Remarks to the Author): with expertise in immunogenic cell death

The manuscript Synergistic effect of tumor chemo-immunotherapy induced by leukocyte-hitchhiking thermal-sensitive micelles by Qi et al shows that E-selectin-modified thermal-sensitive micelles (ES-DSM) co-delivered doxorubicin and SCH 58261 activated by local microwave irradiation-induced hyperthermia and caused T cell-dependent anticancer immunity and tumor reduction.

The authors show that ES-DSM together with hyperthermia cause immunogenic cancer cell death, dendritic cell maturation and T cell activation in vitro. In vivo ES-DSM caused 4T1 lung tumor growth reduction of pulmonary metastasis which was accompanied by an increase in the CTL over Treg ratio. Finally the authors demonstrate generation of immunological memory upon treatment with ES-DSM and hyperthermia.

The manuscript is well written and the methodology is technically sound, nevertheless the manuscript would benefit from additional linguistic editing and more experimental details in both the material and methods section as well as the figure legends.

Response: Thanks very much for your comment. The language of this manuscript had been edited by highly qualified native English-speaking editors at AJE before submission, and the editing certificate was enclosed. However, we still tried our best to further optimize the language, hoping to meet your standards. We also detailed the Methods section as well as the figure legends in the revised version. Parts of the experimental methods were provided in the Supplementary Information.

The authors should explain limited viability of negative controls in Figure 2 e.

Response: Thanks very much for your careful review. We supposed that the limited viability of negative controls in Figure 2e might be due to the poor state of 4T1 cells or the mechanical damage during digestion. We therefore repeated the apoptosis detection using healthy cells, and the results are displayed as follows:

Figure 2. (e) The apoptosis results of 4T1 cells after different treatments for 24 h with or without hyperthermia detected by flow cytometry. (f) The apoptosis rate of 4T1 cells was calculated based on e).

The DAMP release is not well explored.

In vitro the authors should investigate calreticulin (CALR) exposure (not the expression).

ATP is commonly measured by luciferase conversion assays yet experimental details

on how the authors conducted this assay (ELISA) are missing.

In vivo the authors show an increase in the expression of both HMGB1 and CALR whereas a release of HMGB1 and an exposure of CALR would be expected.

Response: Thanks very much for your critical comment about the DAMP exposure.

In the in vitro experiment, the exposure of calreticulin were investigated, rather than the expression, and the results were shown as Figure 2h. But maybe we didn't describe the results very clearly in the original manuscript and misdescribed "exposure" as "expression" in the methodology section. So we complemented the description of the results of calreticulin exposure (last paragraph on page 7), and the mistakes were corrected in the revised manuscript.

In the original manuscript, the concentration of ATP in the supernatant was detected by a commercial ELISA kit (Meimian industrial Co., Ltd., Jiangsu, China), and this experiment was carried out according to the manufacturer's instructions. But we still repeated the ATP detection by luciferase conversion assay according to your professional and constructive advice. The ATP Assay Kit (Beyotime, China) was applied and results showed the same trend as ELISA. The results of luciferase conversion assay have replaced the ELISA results in the revised version, which is as follows:

Figure 2. (j) ATP secretion was detected by luciferase conversion assay.

As for the in vivo detection, the exposure levels of HMGB1 and CRT in the tumor were observed by immunohistochemistry. Generally, the exposure of HMGB1 in tumor is analyzed by this kind of qualitative method [*Nat Commun.* **2017**, 8(1): 1811; *Nano Lett.* **2020**, 20: 1928-1933], but it's still difficult to distinguish between expressed and released HMGB1. So, in many other published articles, the results of HMGB1 release in tumor tissues were not exhibited when the ICD degree of tumors was examined [*Adv Mater.* **2018**, 30(38): e1803001; *ACS Nano.* **2018**, 12(8): 8633-8645; *ACS Nano.* **2020**, 14(10): 13343-13366]. Therefore, in order to be more

rigorous, we decided not to present the immunohistochemical photographs of HMGB1 exposure levels in the revised manuscript. Besides, we further applied immunofluorescence to observe the exposure of calreticulin in tumors, which is clearer method to observe the calreticulin exposure rather than expression, and the results are as follows:

Figure 6. (j) Immunofluorescence was used to examine the level of CRT in tumor sections at the end of the observation period.

As to show the importance of ICD the authors could use blocking antibody against CALR and ectoATPases against ATP *in vivo* and analyse their effect on tumor growth.

Response: Thank you so much for your constructive suggestion. As suggested, we performed a separate experiment to investigate the contribution of ICD to the tumor immunotherapy. 4T1 tumor-bearing mice were randomly sorted into 3 groups (6 mice per group) to respectively receive one of the following treatments once every 3 days: Saline, ES-DSM+MW, ES-DSM+MW+CD39/anti CRT α , for 4 times of treatment. The anti-CRT antibody (Abcam, USA) (10 μ g/mice per injection) and ecto-ATPase CD39 (R&D System, USA) (1 μ g/mice per injection) were intraperitoneally (i.p.) injected every 3 days to block CRT and metabolize ATP starting on day -3. The tumor volume was monitored every 2 days. As displayed in Figure S20, the tumor growth of mice in the ES-DSM+MW+CD39/anti CRT α group could be arrested within first 11 days, but showed poorer efficacy compared with the group of ES-DSM+MW later. This result indicated that although the drug-loaded micelles combined with microwave radiation (MW) could kill tumor cells during the treatment period, it couldn't activate the immune system effectively through the biomarkers of tumor ICD, leading to a poor antitumor effect.

We further analyzed the numbers of DCs and T cells in tumors by flow cytometry. As shown in Figure S21, the infiltration of DCs (CD11c⁺) in tumors of ES-DSM+MW+CD39/anti CRT α group was significantly decreased due to the deactivation of ATP and CRT. Further, the proportions of CD3⁺CD4⁺ and CD3⁺CD8⁺ T cells in tumors also reduced compared with the ES-DSM+MW group. Therefore, the ICD of tumor was critical during the tumor immunotherapy.

Figure S20. The effect of CD39 and anti-CRT antibody on the tumor inhibition of ES-DSM+MW. **(a)** Schematic of the treatment regimen and **(b)** curves showing tumor volumes of mice after various treatments (n=6).

Figure S21. The effect of CD39 and anti-CRT antibody on the tumor infiltration of immune cells. Infiltrations of **a-b)** CD11c⁺ DC and **c-d)** CD3⁺ CD4⁺ and CD3⁺ CD8⁺ T cells in tumors after different treatments were detected by flow cytometry. Immunofluorescence was used to examine the level of **e)** CD8⁺ T cells and **f)** CRT exposure in tumor sections at the end of the observation period.

Yet another convincing assay to demonstrate immunological memory might be the eradication of subcutaneously implanted tumors and rechallenge with antigenetically identical and different syngeneic cancer cells.

Response: Thank you so much for your kind suggestion. After different treatments, the tumor-bearing mice were rechallenged with antigenetically identical cancer cells (4T1 cells), and the tumor growth curves were displayed in Figure 7f in the original manuscript. The growth of the rechallenged 4T1 tumor in the ES-DSM+MW group was the most inhibited, which benefited from the enhanced immunological memory. In the revised version, after the ES-DSM+MW treatment, tumor-bearing mice were further rechallenged with antigenetically different cells (CT26 cells), and the tumor growth was monitored. As exhibited in Figure S23, the rechallenged CT26 tumor could not be effectively suppressed, probably because memory T cells in vivo cannot rapidly respond to neoantigens.

Figure S23. Curves showing volumes of rechallenged CT26 tumor of mice after different treatments (n = 6).

Reviewer #2 (Remarks to the Author): with expertise in adenosine and cancer immunology

The current manuscript by Qi et al. describes a novel technological advance for the localized delivery of chemo- and immunotherapeutic drugs. Whilst liposome like structures have been developed for the purpose of chemotherapy and SCH58261 before (e.g. PMID 29720380) the novelty of the current approach lies in the nature of its thermal sensitive allowing for its spatial and temporal control. Overall the study is well conducted and introduces novelty to the field but I recommend the following to further improve its significance.

1) Throughout the manuscript in the majority of figures, information is missing on the number of times experiments were performed and how many technical or biological replicates are being presented. This makes it very difficult to assess the robustness of the observations. In addition statistical analysis appears absent from many figures where conclusions have been made based upon the observed differences e.g. Figure 3F-O, 4F-M. This should be remedied by the inclusion of statistical tests and/or additional replicates to ensure statistically meaningful data is presented.

Response: Thank you so much for your careful review. As suggested, we added the numbers of experiments performed and replicates in the figure notes, and the statistical analysis was also supplemented for all figures in the revised version.

2) Further example is Figure 7B, it is unclear if this experiment was performed on n=3 mice only, is there a quantification of this particular experiment?

Response: Thank you so much for your careful review. In Figure 7b, there were 3 mice in each group, and we had added this information in the figure note. The quantification of this experiment was provided in Figure S22 in the revised version.

Figure S22. Quantitative analysis of bioluminescence signals of mice (n=3).

3) Whilst the data conclusively show that DM is superior therapeutically to free dox and DSM is superior to DM the relative therapeutic activity of encapsulated SCH vs free SCH is not formally shown. Whilst this is likely given the pharmacokinetic profile of SCH it would be important to confirm this given it is a central message for the manuscript

Response: Thanks for your suggestion. We didn't show the therapeutic efficacy of free or encapsulated SCH because its limited antitumor effect, and it usually applied in combination with other therapies [*J Clin Invest.* **2017**, 127(3): 929-941; *Cancer Immunol Res.* **2015**, 3(5): 506-517]. But as suggested, we supplemented a batch of animal experiments to examine the effect of SCH. Briefly, 4T1 tumor-bearing mice

were randomly sorted into 3 groups (6 mice per group) to respectively receive one of the following treatments once every 3 days: Saline+MW, SCH+MW, ES-SM+MW, for 4 times of treatment. 0.5mg/kg SCH per dose was used in the treatment and at 24 h post-i.v. injection of the test agents, the microwave (MW) was applied locally for 30 min (8 W). The tumor volume was monitored every 2 days and the tumor growth curves were exhibited in Figure S13. Compared to the group of Saline+MW, Free SCH plus MW exhibited poor efficacy, indicating the limited effect of SCH alone. Importantly, the E-selectin-modified SCH-loaded micelles combined with microwave hyperthermia (ES-SM+MW) showed no significant difference in comparison with the SCH+MW group. Although the micelles could increase the amount of SCH at the tumor site, the limited efficacy of SCH alone led to difficulty in tumor suppression. This part of description had been added in the revised manuscript.

Figure S13. Curves showing tumor volumes of mice after SCH+MW or ES-SM+MW treatments (n=6).

4) In Figure 2e it appears that the DSM and ES-DSM formulations induce apoptosis in the absence of hyperthermia but this does not appear to be reflect in 1f, the quantification of this data. Can this be reconciled?

Response: Thanks very much for your careful review. We reanalyzed the apoptosis results in Figure 2e and found that the cells in the negative control (PBS) exhibited a proportion of apoptosis of about 30%. We supposed that it might be due to the poor state of 4T1 cells or the mechanical damage during digestion. We therefore repeated the apoptosis detection using healthy cells, and the results are displayed as follows:

Figure 2. (e) The apoptosis results of 4T1 cells after different treatments for 24 h with or without hyperthermia detected by flow cytometry. (f) The apoptosis rate of 4T1 cells was calculated based on e).

5) For the in vivo analysis although showing the increased proportion of CD8 and CD4 T cells is suggestive of more T cell infiltrate, it would be more compelling to quantify the number of cells.

Response: Thanks very much for your comment. The flow cytometry was used to quantitatively detect the proportion of CD8 and CD4 cells in the tumor, spleen, and PBMC. Due to the length limitation of the main text, we put the quantified statistical results into the Supplementary Information, as shown in Figure S16 and S17.

6) The paper is quite heavy with acronyms that make it very hard to follow the data in the figures. I would recommend that a key for the formulations is included in each figure (or figure legend) to make this easier to follow.

Response: Thanks very much for your kind suggestion. In order to make it easier for readers to follow the data in figures, we added the explanation of each acronym in the figure notes according to your advice.

7) The Conclusion/ Discussion section is very short and misses an opportunity to place this work in the context of the wider field. Some suggested topics to include in this section is how this approach differs from other encapsulation approaches and other immunotherapies that could benefit from such a localised delivery. For example one application would be agonists of innate immunity STING, TLR agonists etc that have toxicity when given systemically.

Response: Thanks very much for your valuable comment. As suggested, we supplemented more topics in the Conclusion section in the revised version and try to

place this work in the context of the wider field, which is as follows:

“Furthermore, due to the maneuverability of drug loading and thermal sensitive characteristic, the micelles provide more opportunities in the field of drug co-delivery and controlled drug release, while reducing drug leakage during the circulation and avoiding toxic effects. In addition to A2AR antagonists, some other immunotherapeutic drugs, such as antagonists of STING, TLR, PD-1/PD-L1 and other targets, can also be delivered by this kind of smart nano systems, thereby increasing tumor accumulation and decreasing systemic toxicity. The combination of immunotherapy and other therapeutic drugs can also be achieved because the thermal sensitive micelles allow for the co-delivery of multiple drugs to improve therapeutic efficacy. Overall, the designed micelles for drug co-delivery not only eliminates the paradoxes between ICD-induced antitumor immunity and adenosine-mediated immunosuppression proposed in this article, thus improving antitumor efficacy, but also provides an effective strategy for targeted delivery and spatiotemporally controlled release of other drugs.”

Reviewer #3 (Remarks to the Author): with expertise in nanosystems – drug delivery – photothermal therapy

Qi et al. present a massive study that examines a combination photo/chemo/nano/immunotherapy for metastatic cancer. Superficially, the final results appear promising, as significant decreases in tumor volume are reported. But the study is so large that none of the results are described in sufficient detail to be reproduced by others, possibly due to space constraints. Critically, the materials are not sufficiently characterized. The key component of the overall strategy is to attach nanoparticles to leukocytes via their known interactions with E selectin, with the hope that nanoparticles containing doxorubicin and a photosensitizer can be transported, i.e. “hitchhike”, to tumors via tumor infiltrating leukocytes. But the strength of this attachment is not examined, and the methods employed are not logical. For example, intravenous injection is employed, which would require extremely high concentrations of nanoparticles to sufficiently tag enough leukocytes to reach the small percentage that would actually migrate to tumors. Additionally, it is not clear how the locally applied microwave radiation could result in enhanced killing of metastatic sites. Of note, the lung tumor model employed does not explain how the microwave radiation is applied to the animals and why this treatment locally effects sites of metastasis and does not cause release of the micelles throughout the mouse.

Of note, the micelles employed have a high CMC and are likely not sufficiently stable to remain attached to leukocytes as they travel, potentially for days, in vivo before reaching tumors. Importantly, this strategy is not novel and may be fundamentally flawed. Their strategy stems from the seminal work of the Irvine group at MIT that developed nanoparticle “backpacking” technology over a decade ago (DOI: 10.1038/nm.2198), where nanoparticles were carefully attached to T cells for enhanced targeting of tumors. In this previous study, the attachment and duration of attachment were both thoroughly investigated and the T cells were strategically stimulated to home to tumors and sites of metastasis. But in the current manuscript by Qi et al., the use of nanoparticles to attach to E selectin may be fundamentally flawed, as this method may actually serve to block the receptors necessary for the leukocytes to extravasate into tumors. Such a strategy needs much more extensive investigation, yet in the current study, it is not even clear which specific cells are being targeted, as there are numerous immune cells (monocytes, macrophages, DCs, neutrophils, eosinophils, certain T cell subsets... etc) expressing receptors that would bind the E selectin modified nanoparticles, yet these cells and their markers are not reported. For these reasons, rejection of this manuscript is recommended. Further details of some key issues that should be addressed prior to publication are outlined below:

Response: Thanks for your comments, but I believe that the major merits of our work were not fully identified by you. The main idea of this manuscript is to eliminate the paradoxes between ICD-induced antitumor immunity and ADO-mediated immunosuppression, further improve tumor therapeutic efficacy. The ICD inducer, DOX, and the A2A adenosine receptor antagonist, SCH 58261, were applied together to kill tumor cells and improve tumor immune microenvironment. While the thermal sensitive micelle, hitchhiking strategy, as well as microwave hyperthermia, were all the optimization options used to facilitate the antitumor effect. Point-by-point responses to your comments are enclosed as follows.

1) Materials are lacking characterization and justification. Why was this specific polymer used, when there are a wide range of thermosensitive materials available?

Response: Thanks for your comment. The core material we used in this research is p-(AAm-co-AN). As described in a published review [*Adv Drug Deliv Rev.* **2019**, 138: 167-192], this polymer is the first UCST-type drug delivery nanocarrier reported by our group [*Angew Chem Int Ed Engl.* **2015**, 54(10): 3126-3131; *Biomaterials.* **2017**, 131: 36-46; *Nano Lett.* **2019**, 19(8): 4949-4959]. Based on these previous studies,

we'd like to further explore the application of this UCST-type polymer in drug delivery, so the specific material was applied in the current work.

The material we used in this research is NTA-PEG-p-(AAm-co-AN), the synthesis and characterization of the polymer were described in detail in the methods section. The chemical structure was confirmed by $^1\text{H-NMR}$ and the spectra were displayed in Figure 1b and S1. The molecular weight of the polymer was determined to be 14.3 kDa by gel permeation chromatography. The upper critical solution temperature (UCST) of the polymer was detected to be 43°C by turbidity measurements. And the critical micelle concentration (CMC) was determined to be $33.2\ \mu\text{g/mL}$ using fluorescence spectroscopy and pyrene as a probe. All the above characterizations were described in the original manuscript (Figure 1a-d), which can verify the structure and properties of the polymer.

2) The CMC of the micelles reported is relatively high, suggesting that the particles will not be stable in vivo, particularly for the time required for cells to migrate to tumors. Of note, the seminal work by the Irvine group required specially crosslinked liposomes for enhanced stability in order to traffic in vivo.

Response: Thanks for your comment. The CMC of the micelles reported was $33.2\ \mu\text{g/mL}$, which could maintain stable after intravenous injection into mice because the concentration of micelles in blood was much higher than the CMC value. Taking DOX as a reference, the drug loading was $2.7 \pm 0.01\%$ and the administrated dose was $3\ \text{mg/kg}$, and the administrated dose of vehicle was calculated to be approximate $2\ \text{mg}$ per mouse. The blood volume of mice is less than $2\ \text{mL}$, indicating the concentration of micelles in blood is more than $1\ \text{mg/mL}$, which is much higher than $33.2\ \mu\text{g/mL}$. Therefore, the micelles could be theoretically stable after intravenously injection. Moreover, a large number of studies have shown that micelles as drug carriers can achieve tumor targeting in vivo [*Adv Mater.* **2018**, 30(3): 10.1002/adma.201705436; *Nano Lett.* **2019**, 19 (4): 2688-2693; *ACS Nano.* **2018**, 12(3): 2426-2439].

3) The method of attaching E selecting to the nanoparticles is not covalent and appears to rely on a His tag interaction that would not be sufficiently stable in vivo.

Response: Thanks for your comment. The interaction between His-tag and Ni-NTA was employed for in vivo application in several previous studies [*Proc Natl Acad Sci USA.* **2014**, 111(3): 930-955; *J Control Release.* **2016**, 223: 215-223]. These

investigators adopted this interaction to attach E-selectin to the surface of liposomes, which then exhibited good leukocyte targeting ability in the peripheral circulation and remained bound to the surface of leukocytes for 72 hours without internalization, demonstrating the in vivo stability of the interaction between His-tag and Ni-NTA. Besides, as displayed in the results of biodistribution in Figure 5a-b and S11, the E-selectin-modified micelles exhibited better tumor accumulation than unmodified micelles, indicating the E-selectin stably existed on micelles during the systemic circulation.

4) The fundamental strategy may be flawed: Wouldn't this "hitchhiking" method serve to block receptors necessary for leukocytes to enter inflamed tissues?

Response: Thanks for your comment. Although the ES-DSM adhesion to leukocytes was achieved by the interaction of E-selectin and sialylated Lewis oligosaccharide, the number of leukocytes was much larger than that of micelles, so the oligosaccharide on leukocytes would not be completely occupied and the related biological activity could be preserved. Besides, the interaction between E-selectin and oligosaccharide does not affect the chemotaxis and penetration ability of leukocytes, so their ability to enter inflammatory tissue will not be inhibited. As biodistribution results in Figure 5a, E-selectin modified micelles exhibited superior tumor targeting ability, which confirmed that the "hitchhiking" method would not affect leukocyte entry into inflammatory tissue. Importantly, a published work also demonstrated that leukocytes attached with E-selectin modified liposomes could still infiltrate into tumors such as prostate cancer, often characterized by dense stromal barriers, subsequently led to a significant reduction in the overall tumor burden [*J Control Release*. **2016**, 223: 215-223].

5) The stability/kinetics of nanoparticle binding to cell surfaces is not examined or reported. How long do the particles reside on the cell surface? Does receptor turnover result in endocytosis of the micelles over time?

Response: Thanks for your comment. After intravenous injection of E-selectin modified micelles (ES-DSM), the leukocytes in blood were isolated and the fluorescence of DOX on leukocytes was detected by flow cytometry. As displayed in Figure 1l and k, under the shear force of peripheral circulation, the micelles could bind to leukocytes within 2 h, and approximately 30% of leukocytes carried DOX at 24 h post-injection. Further, the photographs taken by confocal microscopy in Figure

Im demonstrated that ES-DSM was located on the surface of leukocytes at 24 h post-injection. Taken together, the E-selectin modified micelles could reside on the cell surface for at least 24 h. Importantly, results of the in vivo biodistribution showed that the micelles could be enriched in tumor site at 24 h post-injection, so, the micelles resided on the leukocyte surface for 24 h was qualified for the therapeutic demand. Moreover, in the previous work, the investigators had demonstrated that the E-selectin modified liposomes could remain on the surface of leukocytes for at least 72 h without being internalized [*J Control Release*. **2016**, 223: 215-223], also showed the stability and maintaining of nanoparticle binding to cell surfaces.

6) Prior backpacking strategies noted that the payload would release from the attached nanoparticles and locally modulate the cells to which they were attached. In fact, this was a key part of the Irvine strategy, which was to locally modulate T cells with cytokines while attached to them (DOI: 10.1038/nm.2198). In this current work, wouldn't locally attaching micelles containing a cytotoxic agent serve to decrease viability of the leukocytes to which they are attached?

Response: Thanks for your comment. In Irvine's work, cytokine-loaded nanoparticles were covalently linked to the surface of T cells, and gradually released cytokines were used to modulate T cells, then the activated T cells could suppress tumor development. However, in our study, leukocytes were only served as carriers of drug-loaded micelles to achieve superior tumor enrichment but did not exert antitumor effects. The micelles reported in this manuscript was thermal-sensitive, with very slow release of the payload at physiological temperature (37°C), while it was accelerated under the thermal condition (43°C). This feature could reduce the drug leakage from micelles during systemic circulation, thus avoiding damage to leukocytes to which they adhere, while releasing the drug immediately at tumor sites under the stimulation of local microwave hyperthermia.

We had evaluated the impact of drug-loaded micelles to leukocytes in the original manuscript. The cytotoxicity of ES-DSM to leukocytes was firstly measured. The leukocytes were exposed to ES-DSM at different DOX concentrations for various time at 37°C, the results in Figure S3b indicated the drug-loaded micelles had little toxicity to leukocytes. Further, the chemotaxis and penetration ability of leukocytes were investigated by transwell migration assay. After incubating with DSM or ES-DSM at a DOX concentration of 37.5 µg/mL for 8 h, the number of leukocytes in lower chamber and transwell percentage were similar to the negative control,

indicating the chemotaxis and penetration ability of leukocytes were not affected by drug-loaded micelles. Simultaneously, DSM and ES-DSM were intravenously injected and the leukocytes in blood were isolated after 24 h, the chemotaxis and penetration ability of the isolated leukocytes were also verified by transwell assay (Figure S3c-e).

Figure S3b. Leukocyte viabilities after exposing to ES-DSM at different DOX concentrations for various time.

Figure S3c-e. (c) Images of leukocytes (LEU) transported in the lower chamber of the transwell system in the presence of CXCL2 and CXCL12. (d) The transwell percentage of LEU after incubation with DSM or ES-DSM in vitro was calculated based on (c). (e) The transwell percentage of LEU isolated from mice after intravenous injection of DSM or ES-DSM in vivo was calculated based on (c).

7) The use of an intravenous injection is likely not a viable strategy, as a massive amount of material will have to be injected in order to sufficiently coat enough leukocytes, the vast majority of which will not go to tumors.

Response: Thanks for your comment. We believe that it is unnecessary to coat so many leukocytes with micelles. The leukocyte-hitchhiking strategy reported in this work aims to increase the accumulation of micelles in tumor site. Thus, the major purpose of this work would be achieved as long as the leukocyte-hitchhiking micelles exhibited superior tumor targeting than plain micelles. As shown in Figure 5a, b and S11, E-selectin modified micelles exhibited less liver accumulation and better tumor targeting ability, and the increase of micelles in tumors was benefited from hitching a ride on leukocytes. Consequently, the leukocyte-hitchhiking strategy had achieved the purpose of better tumor targeting by intravenous injection. Moreover, when drug-loaded nanoparticles were administered by intravenous injection, Irvine's group reported that leveraging T cells as vectors greatly enhanced the quantity of drug that could be delivered to tumors, achieving levels in the tumor that were orders of magnitude greater than that which could be delivered by nanoparticles alone [*Sci Transl Med.* **2015**, 7(291): 291ra94; *Nat Commun.* **2017**, 8(1): 1747].

8) The confocal images in figure, 1i show extensive labeling of leukocytes. But it is not clear what cell type is being shown, nor how these cells were extracted. Methods simply state that leukocytes were extracted, and do not report how now where they were extracted from. This level of labeling is unexpected following an intravenous injection, and it appears that this may have actually been an in vitro experiment and not in vivo as reported.

Response: Thanks for your comment.

The extraction method of leukocytes was described in the original Supplementary Information (Page S4), as follows:

“To obtain the leukocytes, the blood of mice was taken by excising eyeballs and the leukocytes were isolated by the mouse peripheral blood leukocyte separation kit according to the manufacturer's instructions (Solarbio, China).”

More details were provided in the Methods section (Page 22), as follows:

“200 μ L of DSM or ES-DSM (concentrations of DOX and SCH were 300 μ g/mL and 50 μ g/mL, respectively) was injected into the mice via the tail vein, and at 2, 8 and 24 h after injection, the leukocytes of treated mice were isolated by the mouse peripheral blood leukocyte separation kit according to the manufacturer's instructions

(Solarbio, China). The DOX fluorescence on the obtained leukocytes was analyzed by flow cytometry (ACEA NovoCyte, USA) and confocal laser scanning microscope (CLSM) (Leica SP8, Germany).”

Importantly, the photograph of leukocytes in the original Figure 1i were obtained according to the above method, rather than labeling in vitro. Moreover, this methodology referred to a published article [*J Control Release.* **2016**, 223: 215-223], which also exhibited an obvious fluorescence signal of drug on leukocytes extracted from peripheral blood in the Fig. 1.

Figure 1 was reconstructed in the revised version and the original Figure 1i had been moved to Figure 1m, which showed the fluorescence of DOX on leukocytes extracted from peripheral blood 24 h after ES-DSM injection, including neutrophils (Right), monocytes (Left), and lymphocytes (Center), identified by the nuclear morphology characteristic.

Figure 1m. Confocal microscopy images of leukocytes 24 hours after the intravenous injection of ES-DSM. Leukocytes have nuclear morphology characteristic of neutrophils (Right), monocytes (Center), and lymphocytes (Left).

Further, leukocytes were isolated from blood 24 h after ES-DSM injection, and APC-labeled anti CD3 or CD16 antibody (BioLegend, USA) were applied to stain T lymphocytes and neutrophils respectively. The confocal microscopy images (Figure S4) further demonstrated that ES-DSM could bind onto the leukocyte surface.

Figure S4. Confocal microscopy images of T lymphocyte and neutrophil 24 hours after the intravenous injection of ES-DSM.

9) In general, targeted cells are not sufficiently examined for their activity nor type and few markers are reported throughout the manuscript.

Response: Thanks for your comment. As mentioned in point 6, we tested the viability,

chemotaxis, and penetration ability of leukocytes after ES-DSM injection, and the results showed that the ES-DSM had little impact on leukocytes. Besides, as described in point 8, we utilized fluorescence-labeled anti CD3 and CD16 antibodies to identify T lymphocytes and neutrophils, which further supported that ES-DSM could bind onto leukocyte surface.

10) The lung metastasis model is lacking in detail. It is not clear how the microwave radiation was applied. Was a single tumor targeted? How were metastatic niches targeted or prevented?

Response: Thanks for your comment. The establishment and treatment of the lung metastasis model was described in the Methods section in the original manuscript. Initially, the orthotopic breast tumor bearing mice was established by injecting 5×10^5 of 4T1 cells. 6 days later, 1×10^5 of Luc-4T1 cells were injected intravenously. Then, the mice were randomly sorted into 5 groups to receive different treatments. The tested agents were injected intravenously and the microwave (MW) radiation was applied 24 h later (8 W, 30 min). The microwave probe was positioned 1 cm away from the fixed animal and oriented towards the orthotopic breast tumor, rather than the metastatic tumors. The metastatic tumors were recognized and killed by tumor-specific immune system, which was strengthened by the drug-loaded micelles supplemented with microwave radiation.

11) Overall, this manuscript appears to be attempting to increase the impact of the work by combining 3-4 separate manuscripts into 1. This work should be broken down into multiple manuscripts so that each step can be more thoroughly examined and reported. As shown, this manuscript lacks detail required to reproduce the reported results in another lab.

Response: Thanks for your comment. The current manuscript was an improvement and optimization on the basis of our previous work [*Nano Lett.* **2019**, 19(8): 4949-4959], and aimed to solve the contradictions existing in the tumor treatment so as to improve the therapeutic efficacy, rather than piecing together multiple efforts.

First, our previous work demonstrated the enhanced tumor inhibition effect facilitated by drug-loaded sensitive nanoparticles, which could induce immunogenic cell death (ICD) of tumor and activate tumor specific immune responses in vivo. However, the large amount of ATP released during tumor ICD would be metabolized into adenosine, which in turn caused immunosuppression via the adenosine pathway.

So, in current work, we combined ICD inducing drug with the adenosine receptor antagonist SCH to relieve the immunosuppressive phenomenon at the tumor site.

Second, in order to increase tumor accumulation of drugs and reduce adverse effects, a thermal sensitive polymer was applied as the nanocarrier to co-deliver DOX and SCH, which could prevent drug leakage during the systemic circulation while release drugs immediately under the thermal condition at tumor site. Consequently, the adverse effects would be alleviated and the antitumor effect would be enhanced.

Third, artificial nanoparticles are still limited by the short systemic circulation time and poor biological penetration, thus resulting in limited accumulation in the tumor site. Leukocytes are circulating cells in the peripheral blood, which have been reported to be utilized to carry drug-loaded nanoparticles and pass challenging biological barriers to accumulate in tumor sites. We therefore attempted to use leukocytes to increase the enrichment of nanoparticles at the tumor site.

Taken together, all efforts have been made to increase the accumulation of nanoparticles in tumor site, achieve spatiotemporally controllable drug release, relieve tumor immunosuppression, and enhance tumor therapeutic efficacy. Each component is critical to the final outcome, and the necessary characterization results have been presented in the manuscript, so we don't think this work can be broken down into multiple manuscripts. To note that, compared to the micelles alone, leukocytes as delivery vehicles for micelles could improve the accumulation of drugs at the tumor site, which meant the main purpose of the current study had been realized. Although the kinetics of nanoparticle binding to cell surfaces is also important, it's not the focus of this work. Besides, we have provided the methodology in as much detail as possible for reference and repetition by others. If, by any chance, any detail was missing, help us to point it out for further improvement.

12) This work also lacks novelty, as this “hitchhiking” strategy is highly dependent on the work of the Irvine group’s nanoparticle “backpacking” technology. Yet the extensive work by the Irvine group as well as several other labs (some of whom have attempted in vivo instead of ex vivo cell tagging) that have adopted this backpacking strategy have not been cited in this manuscript. Authors should thoroughly examine and summarize this prior work, which would provide insight into their strategy and the necessary characterization methods that should be employed for validation.

Response: Thanks for your comment. The “hitchhiking” strategy in our work, which aims to increase the accumulation of drug at tumor sites, is a tool to overcome

biological barriers and achieve tumor targeting, and has been studied by many researchers. In addition to T cells, several other kinds of circulating cells, including neutrophils, NK cells, and erythrocytes, have been also applied in this kind of “hitchhiking” strategy. Therefore, this strategy is only one of the innovations of this manuscript and is an adjunct to improve the antitumor efficacy of drug-loaded micelles. Besides, as suggested, we also cited more references in the Introduction section to improve the manuscript (references 36-39).

Reviewers' Comments:

Reviewer #1:

Remarks to the Author:

The authors have convincingly responded to all questions raised during the initial revision. The quality of the data as well as their presentation have been improved.

Reviewer #2:

Remarks to the Author:

I commend the authors for their careful consideration of the points raised by myself and the other reviewers. I have no further major concerns.

Minor points are

1) On line 370 when the new experiment pertaining to the use of SCH is discussed that the figure s13 is referenced here. The figure is mentioned a few lines prior but it is not immediately obvious to a reader where this data is presented.

2) On line 501 I do not agree/ I do not understand with the authors description of the lack of memory recall response against antigenically different CT26 cells. Isn't the point that the neoantigens against which the therapy induces an immune response expressed in 4T1 cells are no longer relevant in this second tumor model. If memory cells against neoantigens expressed in CT26 cells were present, I would fully expect that the mice would be protected.

Reviewer #3:

Remarks to the Author:

Authors have sufficiently addressed my comments. Publication of the manuscript in its current form is recommended.

REVIEWERS' COMMENTS

Reviewer #1 (Remarks to the Author):

The authors have convincingly responded to all questions raised during the initial revision. The quality of the data as well as their presentation have been improved.

Response: Thank you very much. We are so delighted to receive your positive comment.

Reviewer #2 (Remarks to the Author):

I commend the authors for their careful consideration of the points raised by myself and the other reviewers. I have no further major concerns.

Minor points are

1) On line 370 when the new experiment pertaining to the use of SCH is discussed that the figure s13 is referenced here. The figure is mentioned a few lines prior but it is not immediately obvious to a reader where this data is presented.

Response: Thanks so much for your comment. The Figure S13 has been moved to the main manuscript (Fig. 6e) for the convenience of the reader.

2) On line 501 I do not agree/ I do not understand with the authors description of the lack of memory recall response against antigenically different CT26 cells. Isn't the point that the neoantigens against which the therapy induces an immune response expressed in 4T1 cells are no longer relevant in this second tumor model. If memory cells against neoantigens expressed in CT26 cells were present, I would fully expect that the mice would be protected.

Response: Thanks so much for your careful review. Our description of this result is not appropriate. In 4T1 tumor bearing mice, the immune memory function against 4T1 would be activated after the ES-DSM+MW treatment. When the antigenically different CT26 cells were inoculated as the second tumor model, the immune memory function against 4T1 was no longer relevant to the second CT26 tumor. So the rechallenged CT26 tumor could not be effectively suppressed. This description has been provided in the manuscript.

Reviewer #3 (Remarks to the Author):

Authors have sufficiently addressed my comments. Publication of the manuscript in its current form is recommended.

Response: Thank you very much. We are so delighted to receive your positive comment.